# Decoding reward–curiosity conflict in decision-making from irrational behaviors

**Yuki Konaka** [1] **& Honda Naoki** [1,2,3,4] ✉

Humans and animals are not always rational. They not only rationally exploit rewards but also explore an environment owing to their curiosity. However, the mechanism of such curiosity-driven irrational behavior is largely unknown. Here, we developed a decision-making model for a two-choice task based on the free energy principle, which is a theory integrating recognition and action selection. The model describes irrational behaviors depending on the curiosity level. We also proposed a machine learning method to decode temporal curiosity from behavioral data. By applying it to rat behavioral data, we found that the rat had negative curiosity, reflecting conservative selection sticking to more certain options and that the level of curiosity was upregulated by the expected future information obtained from an uncertain environment. Our decoding approach can be a fundamental tool for identifying the neural basis for reward–curiosity conflicts. Furthermore, it could be effective in diagnosing mental disorders.

Animals and humans perceive the external world through their sensory systems and make decisions accordingly[1,2]. Generally, they cannot make optimal decisions because of the uncertainty of the environment as well as the limited computational capacity of the brain and time constraints associated with decision-making[3]. In fact, they perform irrational actions. For example, people play lotteries and gamble despite low reward expectations. In this case, they face a dilemma between low expected reward and curiosity regarding whether a reward will be acquired. Thus, understanding how animals control the balance between reward and curiosity is important for clarifying the whole decision-making process. However, a method is yet to be established for quantifying the reward–curiosity balance has yet been established. In this study, we developed a machine learning method to decode the time series of the reward–curiosity balance from animal behavioral data.

Some irrational behaviors emerge because of the strength of curiosity[4,5]. For example, conservative individuals avoid uncertainty and prefer to select an action that leads to predictable outcomes. Conversely, inquisitive individuals strongly desire to know the environment rather than rewards and prefer to select an action that leads to unpredictable outcomes. Too conservative and inquisitive natures can be interpreted as autism spectrum disorder and attention deficit hyperactivity disorder, patients with which are known to substantially avoid and seek novel information, respectively[6–14]. Rational individuals fall midway between these two extremes. In an ambiguous environment, they select an action to efficiently understand the environment, and if the environment becomes clear, they select an action to efficiently exploit the rewards. Therefore, curiosity has a major impact on behavioral patterns, and it is believed that animals control the balance between reward and curiosity in a context-dependent manner.

Decision-making has been modeled primarily by reinforcement learning (RL), which is a theory for describing reward-seeking adaptive behavior in which animals not only exploit rewards but also explore the environment. In RL, explorative behavior was addressed by a passive, random choice of action[15]. However, animals actively explore the environment by selecting actions that minimize the uncertainty of the environment given their curiosity.

Recently, the free energy principle (FEP) was proposed by Karl Friston under the Bayesian brain hypothesis, in which the brain optimally recognizes the outside world according to Bayesian estimation[16–18]. The FEP addresses not only the recognition of the external world but also the information-seeking action selection, which minimizes the uncertainty of the recognition of the external world, known as "active

[1]Laboratory of Data-Driven Biology, Graduate School of Integrated Sciences for Life, Hiroshima University, Hiroshima, Japan. [2]Kansei-Brain Informatics Group, Center for Brain, Mind and Kansei Sciences Research, Hiroshima University, Hiroshima, Japan. [3]Theoretical Biology Research Group, Exploratory Research Center on Life and Living Systems, National Institutes of Natural Sciences, Okazaki, Japan. [4]Laboratory of Theoretical Biology, Graduate School of Biostudies, Kyoto University, Kyoto, Japan. ✉e-mail: nhonda@hiroshima-u.ac.jp

inference"[19–21]. Furthermore, FEP proposed a score of action, called expected free energy, which consists of the expected reward and curiosity with the same unit[22–27]. Thus, action selection can be formulated by maximizing both reward and curiosity. Note that curiosity can be regarded as information gain, that is, the extent to which we expect our recognition to be updated by the new observation through the action. However, FEP assumes that the weighting of rewards and curiosity is always even and constant. Although a previous FEP study modeled active inference in a two-choice task, it assumed a constant intensity of curiosity and thus could not treat actual animal behaviors in which the weights of rewards and curiosity are expected to change over time[28]. Hence, conventional theories such as RL and FEP are limited in describing the conflict between reward and curiosity.

Identifying the temporal variability of curiosity is important for future clarifying the neural mechanisms of the reward and curiosity conflicts in decision-making. Many FEP studies have been devoted to the construction of theory, assuming that the decision-making processes of animals are Bayes optimal. Thus, there was not even the idea that animals irrationally make decisions depending on the reward and curiosity conflicts. For this reason, a method to decode the temporal balance between reward and curiosity from behavioral data is yet to be established. Such a method would enable us to analyze neural correlates with the temporal variability of curiosity, and consequently, it would help us clarify how the brain controls the balance of reward and curiosity in a context-dependent manner.

In this study, we extended FEP by incorporating a meta-parameter that controls the conflict dynamics between reward and curiosity, called the reward–curiosity decision-making (ReCU) model. The ReCU model can exhibit various behavioral patterns, such as greedy behavior toward reward, information-seeking behaviors with high curiosity and conservative behaviors avoiding uncertainty. Moreover, we developed a machine learning method called the inverse FEP (iFEP) method to estimate the internal variables of decision-making information processing. Applying the iFEP method to a behavioral time series in a two-choice task, we successfully estimated the internal variables, such as variations in curiosity, recognition of reward availability and its confidence.

## Results

### Decision-making with the reward–curiosity dilemma
Animals perceive the environment by inferring causes such as reward availability from observation, and then they make decisions based on their own inferences. In this study, we developed an ReCU model of a decision-making agent facing a dilemma between reward and curiosity in a two-choice task, wherein the agent selects either of two choices associated with the same rewards but with different reward probabilities (Fig. 1a). If the agent aims to maximize cumulative rewards, the agent must select an option with a higher reward probability. However, in animal behavioral experiments, even after they learned which option was more associated with a reward, they did not exclusively select the best choice, but also often selected the option with a smaller reward probability, which seems unreasonable.

Here, we hypothesized the following: Animals assume that the reward probability for each option might fluctuate over time, and therefore, the continuous selection of one option decreases the confidence of the reward probability estimation for the other option. Thus, they become curious about the ambiguous option even with a smaller reward probability, and so selecting the ambiguous option is reasonable for increasing the confidence of the estimation for both options. Therefore, we considered that the agent should make decisions driven by reward and curiosity in a situation-dependent manner.

### ReCU model
In the ReCU model, we divided information processing in the brain into two processes. In the first process, the agent updates the recognition of the reward probability of each option (Fig. 1a, process 1). In the second

process, the agent selects an action based on the current recognition and curiosity (Fig. 1a, process 2). The agent repeats these two processes in the two-choice task.

We modeled the first process by sequential Bayesian estimation, under the assumption that reward probabilities latently fluctuate in time (Fig. 1b). The agent updates the belief about the reward probabilities, which are expressed as estimation distributions, in response to actions and consequence reward observations (Fig. 1c). We derived the equations of the belief update (Fig. 1d and Methods). We modeled the second process by action selection based on two kinds of motivations: the desires to maximize the reward and to gain the information from the environment (Fig. 1e). This sum, called 'expected net utility' in this study, can be expressed by

$$U_t(a_{t+1}) = E[\text{Reward}_{t+1}] + c_t \cdot E[\text{Info}_{t+1}], \quad (1)$$

where a and t indicate the action and trial index, respectively, and E[x] denotes the expectation value of x based on current recognition. The first and second terms represent expectations of reward and information derived from a new observation, respectively, for the next action $a_{t+1}$ based on current recognition. $c_t$ denotes a meta-parameter describing the intensity of curiosity, which weighs the expected information gain (see Methods for detail). We assumed an irrational mental conflict as $c_t$ varies over time (Fig. 1f). In decision-making, the agents prefer to select action $a_{t+1}$ with the higher expected net utility, in which the action is selected probabilistically following a sigmoidal function

$$P(a_{t+1}) = \frac{1}{1 + \exp(-\beta \Delta U_t)}, \quad (2)$$

where $\Delta U_t = U_t(a_t + 1) - U_t(\overline{a_{t+1}})$, and $\beta$ denotes the inverse temperature controlling the randomness of action selection[22,29,30].

### Recognition and decision-making in the simulation
To validate our model, we performed simulations with constant curiosity $c_t = 1$ for two cases. In the first case, where reward probabilities were constant and different between the two options (Fig. 2a), the agent preferred to select the option with the higher reward probability (Fig. 2b). The recognized reward probabilities converged to ground truths, indicating that the agent accurately recognized the reward probabilities (Fig. 2c). The recognition confidence changed over time depending on the behavior in each trial; the confidence of the selected option increased with information from the observation, whereas that of the unselected option decreased because of the agent's assumption regarding the fluctuation in reward probabilities (Fig. 2d). Similarly, the expected information gain of an option increased and decreased when that option was selected and unselected, respectively. Thus, the expected information gain was lower for the option with higher confidence (Fig. 2e). The expected reward followed the recognized reward probability (Fig. 2f). Initially, decreasing expected information gain and increasing expected reward eventually cross at some number of trials, which correspond to switching between information exploration and reward acquisition (Supplementary Fig. 1a). These two factors are negatively correlated, indicating a trade-off relationship (Supplementary Fig. 1b). The expected net utility, which is the sum of the expected information gain and reward, represents the value of each selection (Fig. 2g), resulting in the agent's preferentially selecting the option with the higher expected net utility.

In the second case, we assumed a dynamic environment with a time-dependent reward probability (Fig. 2h). In the simulation, the agent adaptively changed its recognition of the reward probability following the change in the true reward probability, and selected the option with the higher estimated reward probability (Fig. 2i,j). The confidence was affected by the uncertainty of the reward probability at each time; the confidence was high where the reward probabilities

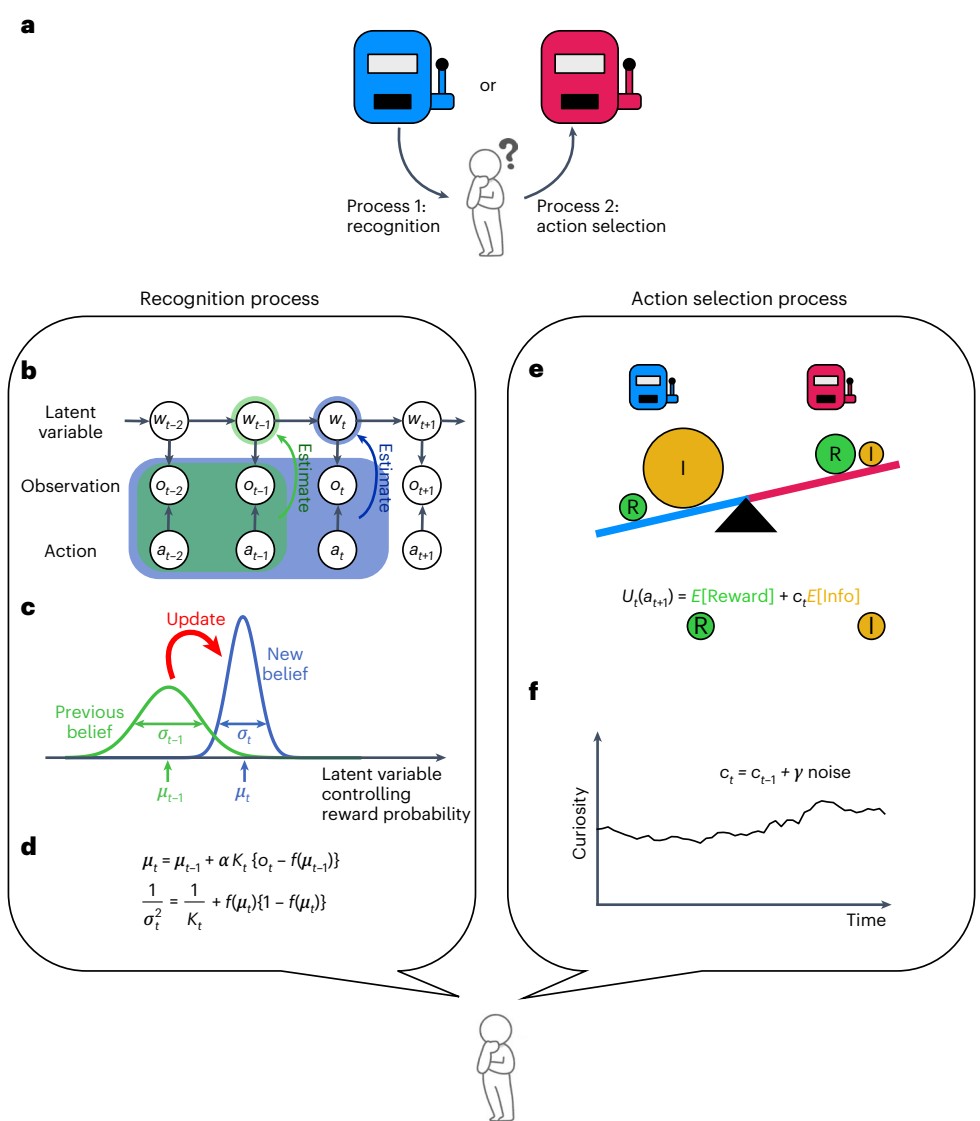

**Fig. 1 | Decision-making model for the two-choice task with reward–curiosity dilemma. a**, Decision-making in the two-choice task. Reward is provided at different probabilities for each option. The agent does not know those probabilities. Through repeated trial and error, the agent recognizes the world by inferring the latent reward probability of each option, and decides to choose the next action, that is, option, based on its own inference. **b**, Sequential Bayesian estimation as a recognition process. The agent assumes that the reward probabilities change over time owing to the fluctuation in the latent variable controlling reward probability. **c**, Belief updating. The agent recognizes the latent variable as a probability distribution. **d**, The update rule of the mean and variance of the estimation distribution for each option. $\alpha$, $K_t$ and $f(\mu_t)$ indicate the learning rate, Kalman gain, and the prediction of the reward probability, respectively. The second term in both equations disappears if the option is not selected. **e**, The action selection process by the agent. The agent evaluates the expected net utility $U_t(a_{t+1})$ of each action using the weighted sum of the expected reward and information gain, as shown in the equation. The agent compares the expected net utilities for both actions and prefers the option with the larger expected net utility. **f**, Time-dependent curiosity. The intensity of curiosity changes over time owing to the fluctuation of $c_t$.

were near-deterministic, around 1 and 0, but low where the reward probabilities were uncertain, i.e., approximately 0.5 (Fig. 2k). The expected information gain of an option was negatively correlated with the confidence of the option (Fig. 2l), suggesting that the agent was curious about the uncertain option. The expected reward just varied depending on the recognized reward probability (Fig. 2m). In this case, we also observed a switching between information exploration and reward acquisition at the initial phase (Supplementary Fig. 1c). In contrast to the above case, the expected reward and expected information gain did not show clearly linear correlation because of the dynamic environment (Supplementary Fig. 1d). The expected net utility changed similarly to the expected reward; however, the difference between the left and right options was less pronounced because of

curiosity (Fig. 2n). These two demonstrations indicate that our model can represent the process of cognition and decision-making based on reward and curiosity.

### Discrimination of passive and curiosity-dependent behaviors
The behavioral difference based on curiosity is interesting. The expected net utility can be rewritten as

$$\Delta U_t = \Delta E\left[\text{Reward}_{t+1}\right] + c_t \cdot \Delta E\left[\text{Info}_{t+1}\right], \qquad (3)$$

where the first and second terms represent the differences between the expected reward and information gain of two options, respectively. Thus, the agent is decided based on the balance between $\Delta E\left[\text{Reward}_{t+1}\right]$

and $\Delta E[\text{Info}_{t+1}]$. Here, we created a diagram visualizing the selected actions in the space of $\Delta E[\text{Reward}_{t+1}]$ and $\Delta E[\text{Info}_{t+1}]$. Depending on the intensity of curiosity $c_t$, the left and right actions can be separated by a boundary of $\Delta E[\text{Reward}_{t+1}] + c_t \cdot \Delta E[\text{Info}_{t+1}] = 0$ (Fig. 2o–q). The agent with $c_t = 0$ selected an option based only on $\Delta E[\text{Reward}_{t+1}]$ (Fig. 2p). In contrast, if $c_t$ was a nonzero value, the boundary leaned to a different direction depending on the positive and negative values of $c_t$ (Fig. 2o,q). These results indicate that passive choice ($c_t = 0$) and curiosity-dependent choice ($c_t \neq 0$) can be discriminated based on the distributed pattern of selected actions in the space of $\Delta E[\text{Reward}_{t+1}]$ and $\Delta E[\text{Info}_{t+1}]$.

## Curiosity-dependent irrational behaviors

We examined how behavioral patterns are regulated by the intensity of curiosity and the degree of reward seeking (Fig. 3). In a scenario where the reward probabilities were zero for the left and 0.5 for the right (Fig. 3a), we simulated a model by varying the meta-parameters $c$ and $P_o$, which is a control parameter of the reward amount (Fig. 3b and Methods). When the agent strongly desired the reward ($P_o = 0.99$) with no curiosity ($c = 0$), the agent preferred the right option with a higher reward probability (Fig. 3c, point a). If the agent had no desire for a reward ($P_o = 0.5$) with high curiosity ($c = 0$), the agent preferred the option with a higher reward probability (Fig. 3c, point b). Although this behavior seems to be rational at first glance, the agent did not seek the reward, but rather sought the information (i.e., belief update) driven by curiosity, which resulted in a preference for the uncertain option. When the agent has negative curiosity ($c = -10$), the agent continuously selected either of the two options depending on the first selection (Fig. 3c, point c). In this behavior, the agent conservatively selected the more certain option, as patients with autism spectrum disorder irrationally avoid new information and repeat the same choices[6–11].

In addition, we obtained a nontrivial result in another scenario, where the reward probabilities were 0.5 for the left and 1 for the right (Fig. 3d–f). As in the previous scenario, the agents with a strong desire for the reward ($P_o = 0.99$, nearly equal to 1) preferred the right option with a higher reward probability (Fig. 3f, point a). The agent with no desire for reward ($P_o = 0.5$) and high curiosity ($c = 10$) preferred the left option with a lower reward probability (Fig. 3f, point b). This seemingly irrational behavior was the outcome of focusing on satisfying curiosity and not seeking rewards, which recalls patients with attention deficit hyperactivity disorder irrationally exploring new information[12–14]. In addition, as seen in the previous scenario (Fig. 3c, point c), the agents with negative curiosity ($c = -10$), irrespective of the desire for the reward, exhibited conservative selection (Fig. 3f, point c). In combination, these results clearly indicate that behavioral patterns largely depend on the degree of conflict between reward and curiosity.

## Inverse FEP: Bayesian estimation of the internal state

In the above cases, we assumed a constant balance between reward and curiosity. However, our feelings swing in a context-dependent manner. Although it is important to decipher the temporal swinging of the conflict between reward and curiosity in terms of neuroscience and psychology, it is difficult to quantify the conflict because of its temporal dynamics. Here, we addressed the inverse problem to estimate the internal states of the agent from behavioral data, known as computational phenotyping or meta-Bayesian inference[31–35]. To this end, we developed a machine learning method called iFEP to quantitatively decipher the temporal dynamics of the internal state including the curiosity meta-parameter from behavioral data.

For developing iFEP, we needed to switch the viewpoint from the agent to the observer of the agent, that is, from animals to us. In the state-space model (SSM) from the viewpoint of the agent, we described the sequential recognition of reward probabilities by the agent (Figs. 1b and 4a,b). Conversely, we developed a state-space model from the observer's eye (the observer-SSM) to determine the internal state of the agent, for example, the intensity of curiosity $c_t$, recognition $\mu_{i,t}$ and its confidence $P_{i,t}$ (i.e., inverse of the variance of the estimation distribution) (Fig. 4c). In the observer-SSM, the intensity of curiosity was assumed to change continuously over time, and the agent's recognition of the reward probability was updated by using the equations shown in Fig. 1c following the FEP; however, they were unknown to the observers. In addition, the agent's actions were assumed to be generated depending on the intensity of its curiosity, recognition and confidence, as described in equation (2), but the observers can only monitor the agent's action and the presence of a reward. In iFEP, based on the observer-SSM, we estimate the latent internal state of agent $z$ from observation $x$ in a Bayesian manner as

$$P(z_{1:T}|x_{1:T}) \propto P(x_{1:T}|z_{1:T})P(z_{1:T}), \tag{4}$$

where $z_{1:T} = \{\mu_{i,1:T}, p_{i,1:T}, c_{1:T}\}$, $x_{1:T} = \{a_{1:T}, o_{1:T}\}$, and the subscript $1:T$ indicates steps 1 to $T$. In this Bayesian estimation, a posterior distribution $P(z_{1:T}|x_{1:T})$ represents the observer's recognition of the estimated $z_{1:T}$ given observation $x_{1:T}$ with uncertainty. A prior distribution $P(z_{1:T})$ represents our belief, which is expressed as the ReCU model with the random motion of the curiosity meta-parameter $c$ as

$$c_t = c_{t-1} + \epsilon\zeta_t, \tag{5}$$

where $\zeta_t$ indicates the white standard Gauss noise and $\epsilon$ indicates its noise intensity. The likelihood $P(x_{1:T}|z_{1:T})$ represents the probability that $x_{1:T}$ was observed assuming $z_{1:T}$, which also follows the ReCU model. This Bayesian estimation, namely iFEP, was conducted using a particle filter and Kalman backward algorithm (Methods).

## Validation of iFEP with artificial data

We tested the validity of the iFEP method by applying it to the artificial data generated by the ReCU model. We simulated a model agent with nonconstant curiosity in the two-choice task, where reward probabilities varied temporally. We then demonstrated that iFEP estimated the ground truth of the internal state of the simulated agent, that is, the agent's intensity of curiosity, recognition and confidence (Supplementary Fig. 2). We also confirmed that the estimation performance is robust against the value of $\epsilon$ (Supplementary Fig. 3). Therefore, iFEP is in a position to provide efficient estimators of belief updating to clarify decision-making processing and the accompanying temporal swing in the conflict between reward and curiosity.

---

**Fig. 2 | Simulations of the decision-making model. a**, The two-choice task with constant reward probabilities. **b**, The selection probabilities for the left and right options, plotted as a moving average with window width of 101. **c**, The recognized reward probabilities for the left and right options compared with the ground truths depicted by dashed lines. **d**, The confidences of reward probability recognitions for left the and right options. **e**–**g**, The expected brief updates (**e**), expected reward (**f**) and expected net utility (**g**) for the left and right options. **h**, The two-choice task with constant and temporally varying reward probabilities for the left and right options. **i**–**n**, The same as **b**–**g** with parameter values of

$c = 1$, $P_o = 0.8$, $\alpha = 0.05$, $\beta = 2$ and $\sigma_w = 0.63$. **o**–**q**, The selected options in a space of left–right differences of the expected reward and information gain. The ReCU model was simulated with dynamically changing reward probabilities for different intensities of curiosity: $c = -1$ (**o**), $c = 0$ (**p**) and $c = 3$ (**q**). The reward probabilities were generated by the Ornstein–Uhlenbeck process of $w$ for 1,000 trials: $w_{i,t} = w_{i,t-1} - 0.01w_{i,t-1} + 0.15\xi_t$, where $\xi_t$ indicates the standard Gauss noise. The heatmap represents the probability of action selection in the space (equation (2)).

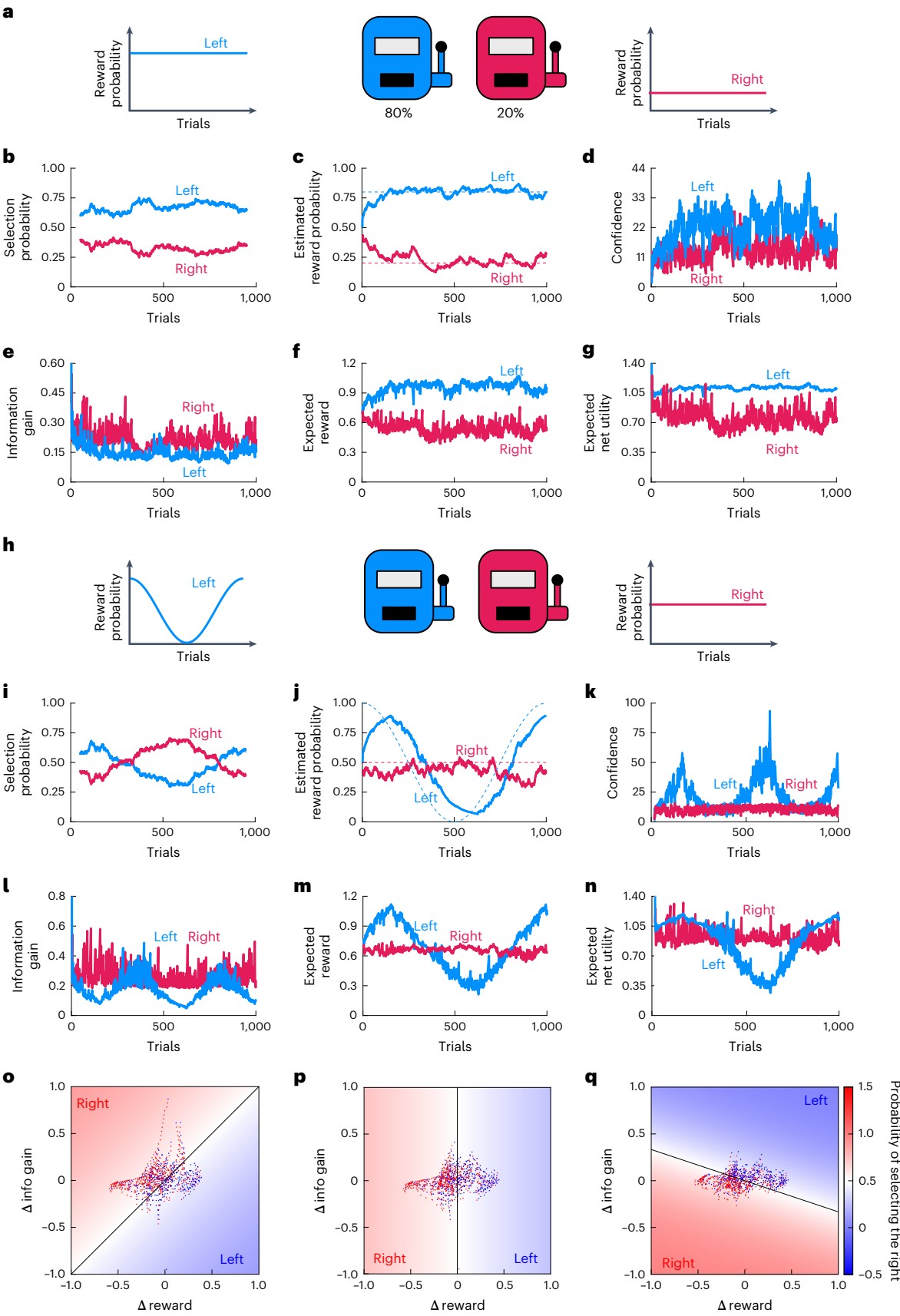

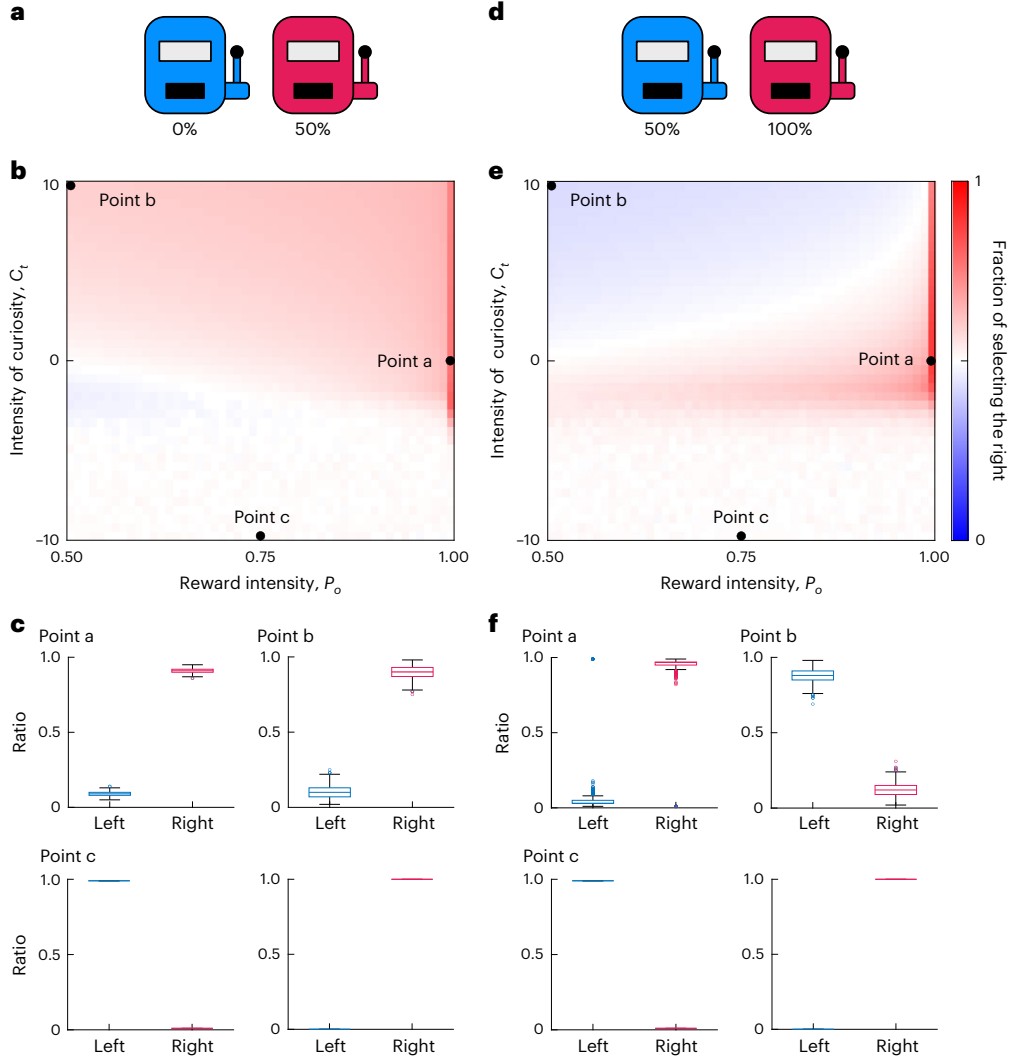

**Fig. 3 | Curiosity-dependent irrational behaviors. a**, The two-choice task with different, constant reward probabilities: 0% for the left and 50% for the right. **b**, The heatmap of the selection probability of the right option as a function of the parameters of curiosity and reward intensity. The probability was obtained empirically by running 1,000 simulations for each set of curiosity and reward. Three representative conditions are indicated by black dots: reward-seeking (point a; $c = 0, P_o = 1$), information-seeking (point b; $c = 10, P_o = 0.5$) and information-avoiding (point c; $c = -10, P_o = 0.75$). **c**, Box plots showing the selection ratios of the right option for the conditions at points a–c, where the

central line indicates the median, the edges are the lower and upper quantiles and the upper and lower whiskers represent the highest and lowest value after excluding outliers, respectively. At point c, the model agent dominantly selected either the right or left option, where the left- and right-dominantly selected simulations occurred 505 and 495 times, respectively. The box plots for point c appear crushed because the data points are too densely packed. **d**–**f**, The same as **a**–**c** for the two-choice task with different, constant reward probabilities: 50% for the left and 100% for the right. At point c in **f**, the left- and right-dominantly selected simulations occurred 489 and 511 times, respectively.

## iFEP-decoded internal state behind rat behaviors

We applied iFEP to actual rat behavioral data from the two-choice task experiment with temporally varying reward probabilities[36] (Fig. 5a). In this experiment, once the reward probabilities were suddenly changed in a discrete manner, the rat slowly adapted to select the option with the higher reward probability (Fig. 5b), suggesting that the rat sequentially updated its recognition of the reward probability. Based on these behavioral data of the rat, iFEP estimated the internal state, that is, the intensity of curiosity, the recognized reward probabilities and their confidence levels (Fig. 5c–e). We found that the rat was not perfectly aware of the true reward probabilities but was able to recognize increases and decreases in reward probability (Fig. 5d,e). We also found that confidence increased with choice and decreased with no choice (Fig. 5f,g).

With iFEP, we can examine whether the rat subjectively assumed fluctuating or constant environments, that is, reward probabilities.

We estimated the degree of fluctuation of the reward probabilities the rat assumes $p_w = 1.785$ (that is, $\sigma_w^2 = 0.560$) from the rat behavioral data. This estimated value implied that the latent variable controlling the reward probabilities showed a random walk with increasing s.d. with trials as $\sqrt{\sigma_w^2 t}$. Compared with a reward probability represented by the sigmoidal of the latent variable, the estimated reward probability can largely change from 0.5 to 0.5 ± 0.4 during only ten trials (Supplementary Fig. 4). Therefore, it was suggested that the rat assumed fluctuating environments and, thus, easily forgot its recognition and lost its confidence.

## Negative curiosity and its dynamics decoded by iFEP

Interestingly, the curiosity held by the rat was estimated to be negative for almost all trials (Fig. 5h). In other words, the rat conservatively preferred certain choices but did not explore uncertain choices.

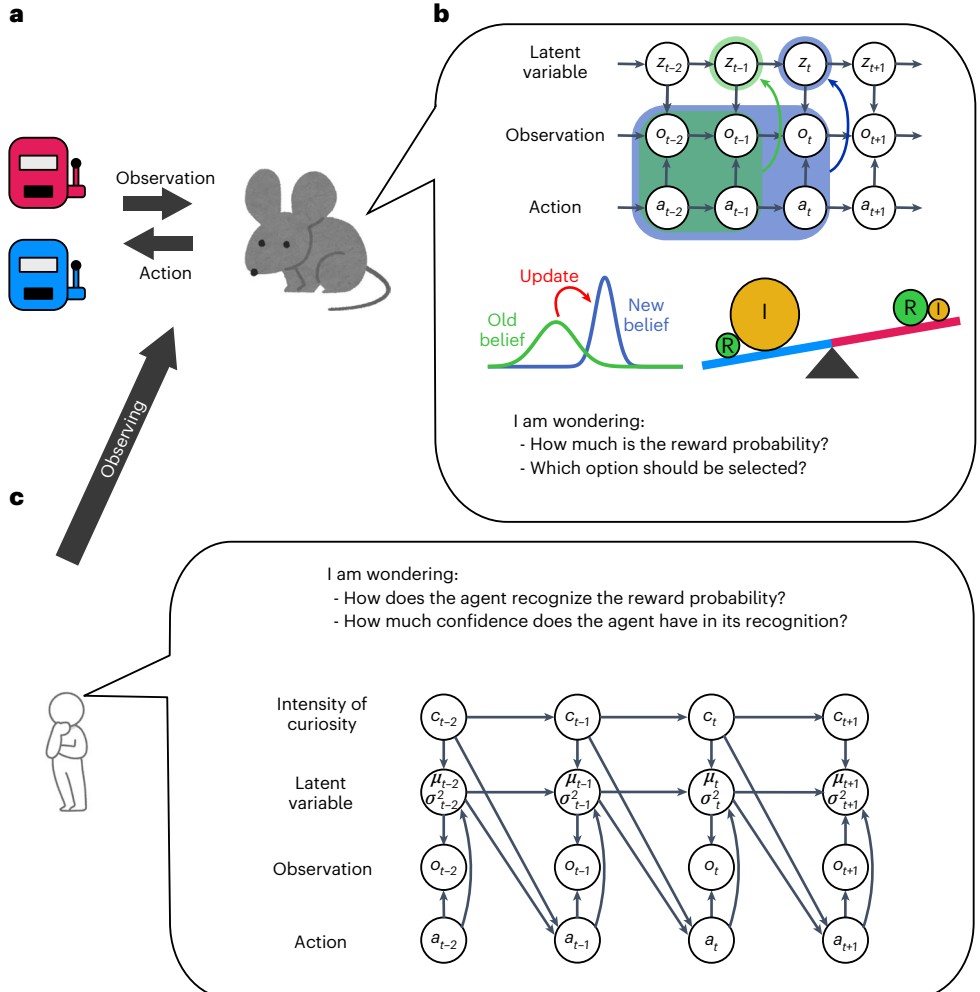

**Fig. 4 | The scheme of iFEP by an observer of a decision-making agent.**
**a**, An agent performing a two-choice task from the observer's perspective.
**b**, The observer's assumption about the agent's decision-making. The agent is
assumed to follow the decision-making model, as described in Fig. 1. **c**, The SSM
of the observer's eye. For the observer, the agent's reward–curiosity conflict,
recognized reward probabilities and their uncertainties are temporally varying
latent variables, whereas the agent's action and the presence/absence of a reward
are observable. The observer estimates the latent internal states of the agent.

This negative curiosity is reasonable for the starved animals because
animals desired to obtain more reward with higher confidence. To
validate the negative curiosity, we visualize the selected action in the
space of $\Delta E\left[\text{Reward}_{t+1}\right]$ and $\Delta E\left[\text{Info}_{t+1}\right]$, as shown in Fig. 2o–q
(Fig. 6a–c). When the estimated $c_t$ was negative, the rat dominantly
selected the left action in a region with positive $\Delta E\left[\text{Reward}_{t+1}\right]$ and
negative $\Delta E\left[\text{Info}_{t+1}\right]$ (Fig. 6a for $c_t < -1.1$; Fig. 6b for $-1.1 \leq c_t < -0.7$),
clearly indicating that the rat has positive subjective reward and nega-
tive curiosity. When the estimated $c_t$ was close to 0, the rat selected
both actions based on $\Delta E\left[\text{Reward}_{t+1}\right]$, independent of $\Delta E\left[\text{Info}_{t+1}\right]$
(Fig. 6c for $-0.7 \leq c_t$). These results clearly supported the expected
net utility with positive weight for the expected reward and time-
dependent weight for the expected information gain. In addition, we
statistically tested negative curiosity ($P < 0.01$ for Monte Carlo testing
in Supplementary Fig. 5).

Further, we noticed an increase in the estimated level of curiosity at
which the reward probabilities changed suddenly (Fig. 5h). This curios-
ity dynamics can be interpreted such that the rat recognized the rule
change and adaptively controlled the extent to which the rat sought
new information. We further examined how the curiosity is regulated
by recognized environmental information. We did not detect the cor-
relation between the estimated curiosity and expected information
gains (Fig. 6d,e). Moreover, we found that the temporal derivative of

the estimated curiosity highly correlated with the sum of the expected
information gains for both options (Fig. 6f,g). These results implied that
the rat actively upregulated the curiosity level when the uncertainty
in the recognized reward probabilities increases, such as,

$$\frac{\mathrm{d}c}{\mathrm{d}t} \propto \sum_i E\left[\text{Info}_i\right]. \tag{6}$$

**Evaluations of alternative models from rat behaviors**
Finally, to further confirm the validity of the ReCU model, we compared
it with other decision-making models based on the rat behavioral data.
As an alternative version of the expected net utility, we introduced the
time-dependent desire for reward as

$$U_t\left(a_{t+1}\right) = d_t \cdot E\left[\text{Reward}_{t+1}\right] + E\left[\text{Info}_{t+1}\right], \tag{7}$$

where $d_t$ denotes a meta-parameter describing subjective reward (see
Methods for details). With this alternative model, the time series of $d_t$
were estimated by iFEP from the rat behavioral data. We found that the
estimation of the subjective reward meta-parameter $d_t$ changed dynam-
ically and sometimes became close to zero when the rat encountered
drastic changes in the reward probabilities (Supplementary Fig. 6),
which indicated that the rat suddenly no longer needed rewards. This

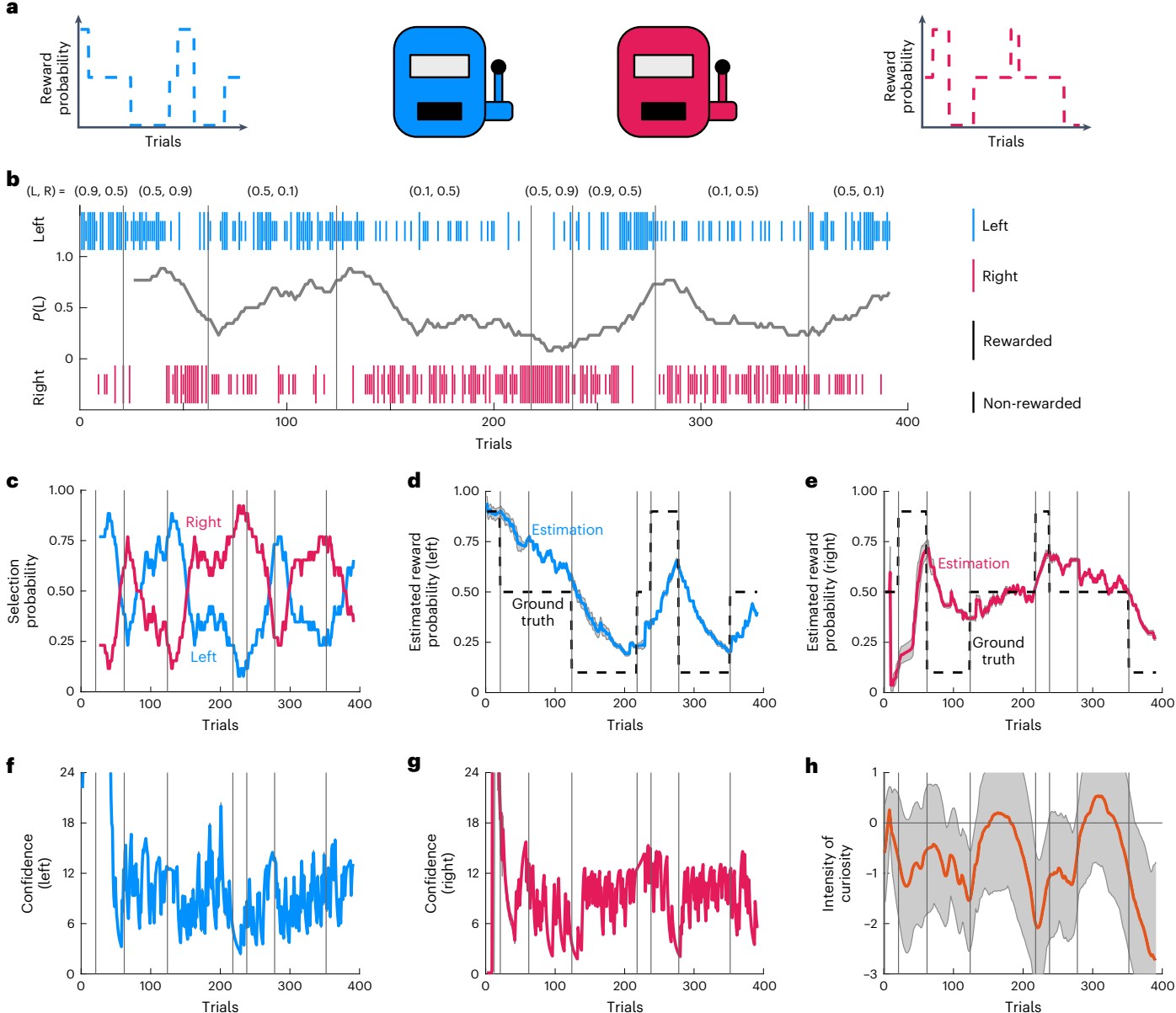

**Fig. 5 | Estimation of the rat's internal state by iFEP. a**, A rat in the two-choice task with temporally switching reward probabilities. **b**, Rat's behaviors from a public dataset at https://groups.oist.jp/ja/ncu/data ref. 5. Vertical lines indicate the selections of left (L) and right (R) options, respectively. The time series indicates the moving average of the selection probability of the left option with 25 window width backward. **c**, The moving average of the selection probabilities for the left and right options. **d,e**, The rat behavior data-driven estimations of agent-recognized reward probabilities for the left (**d**) and right (**e**) options. **f**–**h**, The rat behavior-driven estimations of agent's confidence about recognized reward probabilities for the left (**f**) and right (**g**) options, and the agent's curiosity (**h**). The estimated parameter values were $\alpha = 0.058$, $\beta = 6.991$ and $\sigma_w^2 = 0.560$. The number of particles was 100,000 in the particle filter. Continuous shaded ranges represent the s.d. for all the particles in **d**–**h**.

should be unnatural for animals that were starved before the experimental task to motivate them to obtain food. Thus, the alternative model is not suitable for describing the rat behavior.

Another possible model is Q-learning in the framework of RL, which has been widely used to model the decision-making tasks. Following previous studies[36–38], we introduced the time-dependent inverse temperature $B_t$, which controls the randomness of the action selection; however, it does not lead to information-seeking behavior in the ReCU model. With the Q-learning model, we estimated time series of $B_t$ (Methods). Then, we determined that the $B_t$ decreased when the reward probabilities suddenly changed (Supplementary Fig. 7), meaning that the rat tends to perform a random selection of actions in response to the environmental rule change. Although this dynamic behavior

of the inverse temperature seems reasonable, it is unknown how it is regulated in the Q-learning model. Here, we hypothesized that $B_t$ could be regulated by the uncertainty of our recognition. To probe this, we compared $B_t$ and the expected information gain, where the former was estimated based on Q-learning and the latter was estimated by iFEP in the ReCU model. We found that they are positively correlated (Supplementary Fig. 7) as

$$B_t \propto \sum_i E\left[\mathrm{Info}_{t,i}\right]. \tag{8}$$

Therefore, Q-learning requires a cue regarding the uncertainty of recognition, that is, the expected information gain, which supports the ReCU model for explaining the curiosity-driven behavior.

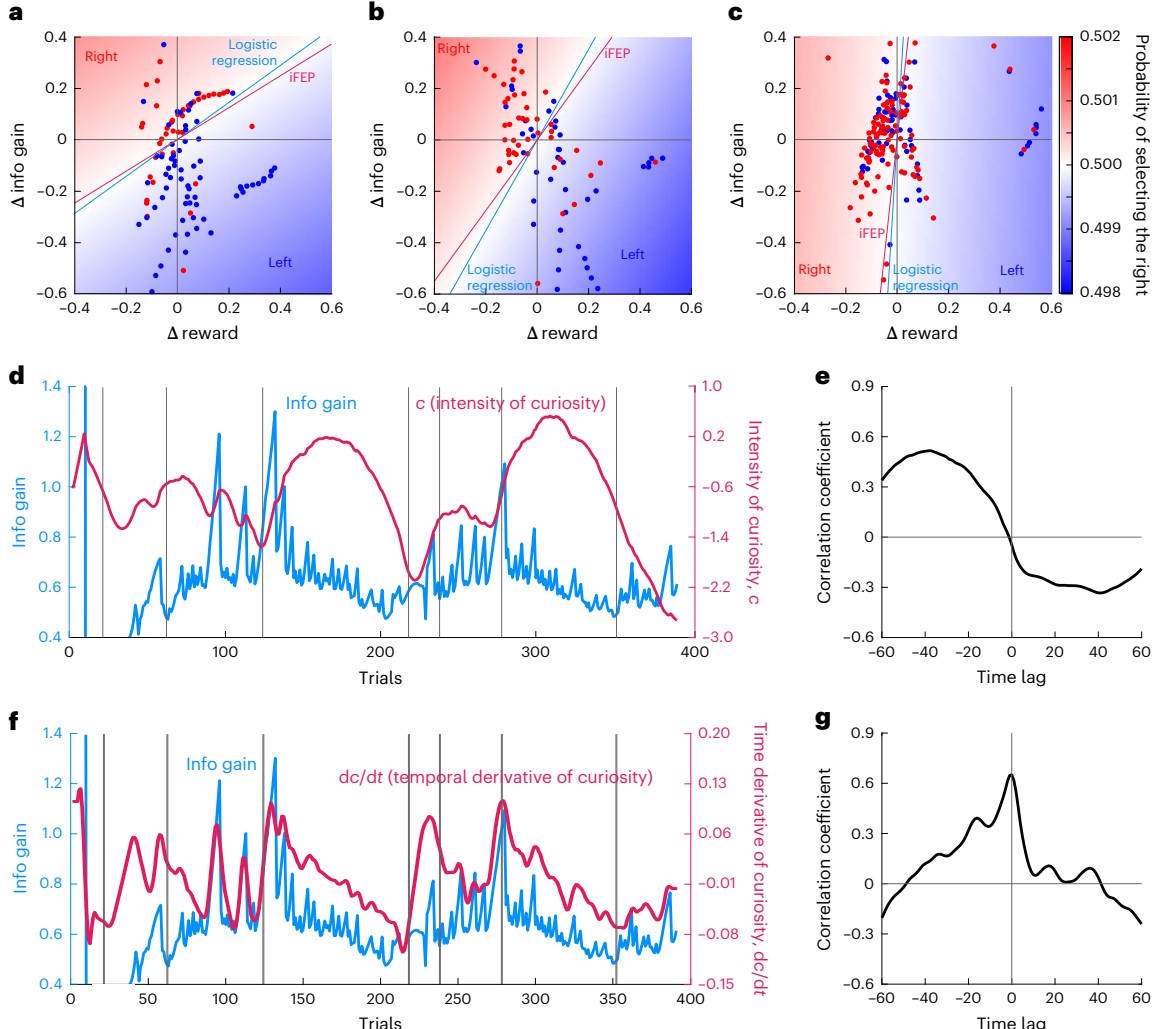

**Fig. 6 | Negative curiosity and its dynamics. a–c**, The selected options in a space of left–right differences of the expected reward and information gain. All actions were separated into three: when the estimated curiosity is negatively large ($c \leq -1.1$, $n = 110$) (**a**), negatively medium ($-1.1 < c \leq -0.7$, $n = 93$) (**b**) and close to 0 ($-0.7 < c$, $n = 187$) (**c**). The left and right options were discriminated by a logistic regression with $P(a_t = 1) = f(w_R \Delta E [\text{Reward}_{t+1}] + w_I \Delta E [\text{Info}_{t+1}])$, where $w_R$ and $w_I$ indicate the weight parameters. Two linear lines indicate discrimination boundaries using the estimated $w_R$ and $w_I$ from this scattered data and using

$w_R = 1$ and $w_I = \sum c_t / N$, which is an average of the estimated curiosity, respectively. the heatmap represents the probability of selecting the left option based on the estimated $w_R$ and $w_I$. **d,e**, Time series of the intensity of curiosity and the sum of the expected information gains for both options (**d**), and their cross-correlation (**e**). **f,g**, Time series of the temporal derivative of curiosity and the sum of the expected information gains for both options (**f**), and their cross-correlation (**g**). The temporal derivative was computed by linear regression within a time window of seven trials.

## Discussion

The advancement realized in this study is the modeling and decoding of mental conflict between reward and curiosity, which is yet to be quantified. The proposed approach can potentially improve our understanding of how mental conflict is regulated and highlight its neural mechanisms in the future by combining neural recording data.

Our decoding approach has some limitations. The iFEP requires long trial behavioral data in the two-choice task because the particle filter needs certain trials for converging the estimation. Thus, the iFEP is not applicable to short behavioral data. In addition, the iFEP assumed gradual change in curiosity in time and cannot follow the pathologically rapid dynamics of curiosity in the estimation process of the particle filter.

Comparing the ReCU model and other models including Q-learning is difficult. In all these models, the action selection is commonly formulated with the sigmoidal function. Thus, any models can be made to fit the observed behaviors, i.e., increase likelihood, by adjusting the time-dependent meta-parameters. Therefore, model selection

based solely on likelihood is not very helpful and determining whether animals use the ReCU model or other models to make decisions seems inherently challenging. However, a potential advantage of the ReCU model is its ability to capture the dynamics of curiosity and the interplay between curiosity and reward-based learning.

There is a related theoretical model, which differs from RL and FEP. Ortega and Braun formulated FEP, which describes irrational decision making[39,40]. Interestingly, their formulation was based on microscopic thermodynamics and the temperature parameters controlled this irrationality. However, the thermodynamics-based FEP did not treat the sequential update of the recognition from the observations.

Finally, it is worth discussing future perspectives of our iFEP approach in medicine. In general, mental diagnosis relies on medical interviews and has not been evaluated quantitatively. Our iFEP method can quantitatively estimate the psychological state of patients based on their behavioral data. For example, patients with social withdrawal, also known as 'Hikikomori,' have no interest in anything. In this case, social withdrawal would be characterized by a negative value of curiosity in

our FEP model. Therefore, the iFEP method could be considered effective for diagnosing mental disorders.

# Methods

## Amount of reward

In the two-choice task, the reward is given as all-or-none; however, its intensity depends on the agent's own feeling as

$$R = 0 \text{ or } \ln \frac{P_o}{1-P_o}, \qquad (9)$$

where $P_o$ represents the desired probability of how much the agent wants the reward and controls the reward intensity felt by the agent (Supplementary Fig. 8). In Friston's formulation, the presence and absence of rewards are given by log preference with natural unit as ln $P_o$ and ln$(1-P_o)$, which take negative values[22,23]. In this equation, the reward is the difference between log preferences to ensure that the reward intensity is positive.

## State space model for reward probability recognition

The agent assumed that the reward is generated probabilistically from the latent cause $w$, and probabilities $\lambda_i$ are represented by

$$\lambda_i = f(w_i), \qquad (10)$$

where $i$ indicates the indices of the options and $f(x) = 1/(1 + e^{-x})$. In addition, the agent assumed an ambiguous environment in which the reward probabilities are fluctuated by a random walk as

$$w_{i,t} = w_{i,t-1} + \sigma_w \xi_{i,t}, \qquad (11)$$

where $t$, $\xi_{i,t}$, and $\sigma_w$ denote the trial of the two-choice task, standard Gaussian noise and the noise intensity, respectively. Thus, the agent assumes an environment expressed by an SSM as

$$P(\mathbf{w}_t|\mathbf{w}_{t-1}) = \mathcal{N}\left(\mathbf{w}_t|\mathbf{w}_{t-1}, \sigma_w^2\mathbf{I}\right), \qquad (12)$$

$$P(o_t|\mathbf{w}_t, \mathbf{a}_t) = \prod_i \left[ f(w_{i,t})^{o_t} \{1-f(w_{i,t})\}^{1-o_t} \right]^{a_{i,t}}, \qquad (13)$$

where $\mathbf{w}_t$ and $o_t$ denote the latent variables controlling the reward probabilities of both options at step $t$ ($\mathbf{w}_t = (w_{1,t}, w_{2,t})^{\mathrm{T}}$) and the observation of the presence of the reward ($o_t \in \{0,1\}$), respectively. Meanwhile, $\mathbf{a}_t$ denotes the agent's action at step $t$, which is represented by a one-hot vector $\left(\mathbf{a}_t \in \left\{(1,0)^{\mathrm{T}}, (0,1)^{\mathrm{T}}\right\}\right)$, $\mathcal{N}(\mathbf{x}|\mu, \Sigma)$ denotes the Gaussian distribution mean $\mu$ and variance $\Sigma$, $\sigma_w^2$ denotes the variance of the transition probability of $\mathbf{w}$, $\mathbf{I}$ denotes an identity matrix, and $f(w_{i,t}) = 1/(1 + e^{-w_{i,t}})$, which represents the probability of the reward of option $i$ at step $t$. The initial distribution of $\mathbf{w}_1$ is given by $P(\mathbf{w}_1) = \mathcal{N}(\mathbf{w}_1|0, \kappa\mathbf{I})$, where $\kappa$ denotes variance.

## FEP for reward probability recognition

We modeled the agent's recognition process of reward probability using sequential Bayesian updating as

$$P(\mathbf{w}_t|o_{1:t}, \mathbf{a}_{1:t}) \propto P(o_t|\mathbf{w}_t, \mathbf{a}_t) \int P(\mathbf{w}_t|\mathbf{w}_{t-1}) P(\mathbf{w}_{t-1}|o_{1:t-1}, \mathbf{a}_{1:t-1}) \, d\mathbf{w}_{t-1}. \quad (14)$$

Because of the non-Gaussian $P(o_t|\mathbf{w}_t, \mathbf{a}_t)$, the posterior distribution of $\mathbf{w}_t$, $P(\mathbf{w}_t|o_{1:t}, \mathbf{a}_{1:t})$, becomes non-Gaussian and cannot be calculated analytically. To avoid this problem, we introduced a simple posterior distribution approximated by a Gaussian distribution:

$$Q(\mathbf{w}_t|\varphi_t) = \mathcal{N}\left(\mathbf{w}_t|\mu_t, \Lambda_t^{-1}\right) \coloneqq P(\mathbf{w}_t|o_{1:t}, \mathbf{a}_{1:t}), \qquad (15)$$

where $\varphi_t = \{\mu_t, \Lambda_t\}$, and $\mu_t$ and $\Lambda_t$ denote the mean and precision, respectively ($\mu_t = (\mu_{1,t}, \mu_{2,t})^{\mathrm{T}}$; $\Lambda_t = \mathrm{diag}(p_{1,t}, p_{2,t})$). $Q(\mathbf{w}_t|\varphi_t)$ denotes the recognition distribution. The model agent aims to update the recognition distribution through $\varphi_t$ at each time step by minimizing the surprise, which is defined by $-\ln P(o_t|o_{1:t-1})$. The surprise can be decomposed as follows:

$$-\ln P(o_t|o_{1:t-1}) = \int Q(\mathbf{w}_t|\varphi_t) \ln \frac{Q(\mathbf{w}_t|\varphi_t)}{P(o_t, \mathbf{w}_t|o_{1:t-1}, \mathbf{a}_{1:t})} \, d\mathbf{w}_t$$

$$-\mathrm{KL}\left[Q(\mathbf{w}_t|\varphi_t)\|P(\mathbf{w}_t|o_{1:t}, \mathbf{a}_{1:t})\right], \qquad (16)$$

where $\mathrm{KL}[q(\mathbf{x})\|p(\mathbf{x})]$ denotes the Kullback–Leibler (KL) divergence between the probability distributions $q(\mathbf{x})$ and $p(\mathbf{x})$. Because of the nonnegativity of KL divergence, the first term is the upper bound of the surprise:

$$F(o_t, \varphi_t) = \int Q(\mathbf{w}_t|\varphi_t) \ln Q(\mathbf{w}_t|\varphi_t) \, d\mathbf{w}_t + \int Q(\mathbf{w}_t|\varphi_t) J(o_t, \mathbf{w}_t) \, d\mathbf{w}_t, \quad (17)$$

which is called the free energy, where $J(o_t, \mathbf{w}_t) = -\ln P(o_t, \mathbf{w}_t|o_{1:t-1}, \mathbf{a}_{1:t})$. The first term of the free energy corresponds to the negative entropy of a Gaussian distribution:

$$F_1 = \int Q(\mathbf{w}_t|\varphi_t) \ln Q(\mathbf{w}_t|\varphi_t) \, d\mathbf{w}_t. \qquad (18)$$

The second term is approximated as

$$F_2 = \int Q(\mathbf{w}_t|\varphi_t) J(o_t, \mathbf{w}_t) \, d\mathbf{w}_t$$

$$\cong \int Q(\mathbf{w}_t|\varphi_t) \left\{ J(o_t, \mu_t) + \frac{\mathrm{d}J}{\mathrm{d}w}(\mathbf{w}_t - \mu_t) + \frac{1}{2}\frac{\mathrm{d}^2J}{\mathrm{d}w^2}(\mathbf{w}_t - \mu_t)^2 \right\} \mathrm{d}\mathbf{w}_t \quad (19)$$

$$= J(o_t, \mathbf{w}_t)|_{w_t=\mu_t} + \frac{1}{2}\frac{\mathrm{d}^2J}{\mathrm{d}w^2}\Big|_{w_t=\mu_t} \Lambda_t^{-1}.$$

Note that $E(o_t, \mathbf{w}_t)$ is expanded by a second-order Taylor series around $\mu_t$. At each time step, the agent updates $\varphi_t$ by minimizing $F(o_t, \varphi_t)$.

## Calculation of free energy

The free energy is derived as follows: $F_1$ simply becomes

$$F_1 = \frac{1}{2}\ln 2\pi p_{1,t}^{-1} + \frac{1}{2}\ln 2\pi p_{2,t}^{-1} + \mathrm{const.} \qquad (20)$$

For computing $F_2$,

$$P(o_t, \mathbf{w}_t|o_{1:t-1}, \mathbf{a}_{1:t}) = P(o_t|\mathbf{w}_t, \mathbf{a}_t) \int P(\mathbf{w}_t|\mathbf{w}_{t-1}) P(\mathbf{w}_{t-1}|o_{1:t-1}, \mathbf{a}_{1:t-1}) \, d\mathbf{w}_{t-1}$$

$$\cong P(o_t|\mathbf{w}_t, \mathbf{a}_t) \int P(\mathbf{w}_t|\mathbf{w}_{t-1}) N\left(\mathbf{w}_{t-1}|\mu_{t-1}, \Lambda_{t-1}^{-1}\right) \, d\mathbf{w}_{t-1}. \qquad (21)$$

In the second line of this equation, we use the approximated recognition distribution as the previous posterior $P(\mathbf{w}_{t-1}|o_{1:t-1}, \mathbf{a}_{1:t-1})$. This equation can be written as

$$P(o_t, \mathbf{w}_t|o_{1:t-1}, \mathbf{a}_{1:t}) \cong \prod_i \left[ f(w_{i,t})^{o_t} \{1-f(w_{i,t})\}^{1-o_t} \right]^{a_{i,t}} N\left(w_{i,t}|\mu_{i,t-1}, p_{i,t}^{-1} + \sigma_w^2\right). \qquad (22)$$

Then

$$E(o_t, \mathbf{w}_t)|_{w_t = \mu_t} = J_1(o_t, \mu_{1,t}) + J_2(o_t, \mu_{2,t}) + \text{const.}, \tag{23}$$

where

$$J_i(o_t, \mu_{i,t}) = a_{i,t}\left[o_t \ln f(\mu_{i,t}) + (1 - o_t)\ln\{1 - f(\mu_{i,t})\}\right] \tag{24}$$

$$-\frac{1}{2}\frac{(\mu_{i,t} - \mu_{i,t-1})^2}{p_{i,t}^{-1} + \sigma_w^2} - \frac{1}{2}\ln\left(p_{i,t}^{-1} + \sigma_w^2\right).$$

Thus, $F_2$ is calculated by substituting this equation into equation (19). Taken together,

$$F(o_t, \varphi_t) = \sum_i \left\{ J_i(o_t, \mu_{i,t}) + \frac{1}{2}\left.\frac{\mathrm{d}^2 J_i}{\mathrm{d}w_{i,t}^2}\right|_{w_{i,t} = \mu_{i,t}} p_{i,t}^{-1} + \frac{1}{2}\ln 2\pi p_{i,t}^{-1} \right\}. \tag{25}$$

### Sequential updating of the agent's recognition

The updating rule for $\varphi_t$ was derived by minimizing the free energy. The optimized $p_{i,t}$ can be computed by $\partial F/\partial p_{i,t}^{-1} = 0$, which leads to

$$p_{i,t} = \left.\frac{\mathrm{d}^2 J_i}{\mathrm{d}w_{i,t}^2}\right|_{w_{i,t} = \mu_{i,t}}. \tag{26}$$

By substituting $p_{i,t}$ into equation (25), the second term in the summation becomes constant, irrespective of $\mu_{i,t}$. Thus, $\mu_{i,t}$ is updated by minimizing only the first term as

$$\mu_{i,t} = \mu_{i,t-1} - \alpha\delta_{i,a}\left.\frac{\partial J_i}{\partial \mu_{i,t}}\right|_{\mu_{i,t} = \mu_{i,t-1}}, \tag{27}$$

where $\alpha$ is the learning rate. These two equations finally lead to

$$\mu_{i,t} = \mu_{i,t-1} + \alpha K_{i,t}\left(o_t - f(\mu_{i,t-1})\right), \tag{28}$$

$$p_{i,t} = K_{i,t}^{-1} + f(\mu_{i,t})\left(1 - f(\mu_{i,t})\right), \tag{29}$$

where $K_{i,t} = (p_{i,t-1} + \sigma_w^{-2})/(p_{i,t-1}\sigma_w^{-2})$, which is called the Kalman gain. If option $i$ was not selected, the second terms in both equations will vanish, which results in belief $\mu_{i,t}$ staying the same, while its precision decreases (that is, $p_{i,t+1} < p_{i,t}$). If it is selected, the belief is updated by the prediction error (that is, $o_t - f(\mu_{i,t})$), and its precision is improved. The confidence of the recognized reward probability should be evaluated not in $w_{i,t}$ space but in $\lambda_{i,t}$ space; hence, the confidence is defined by $\gamma_{i,t} = p_{i,t}/f'(\mu_{i,t})^2$.

### Expected net utility

The expected net utility is described by

$$U_t(\mathbf{a}_{t+1}) = c_t \cdot E_{P(o_{t+1}|\mathbf{a}_{t+1})}\left[\text{KL}\left[Q(\mathbf{w}_{t+1}|o_{t+1}, \mathbf{a}_{t+1})\|Q(\mathbf{w}_{t+1}|\mathbf{a}_{t+1})\right]\right] \tag{30}$$
$$+ E_{P(o_{t+1}|\mathbf{a}_{t+1})}\left[R(o_{t+1})\right],$$

where $R(o_{t+1}) = o_{t+1}\ln(P_o/(1 - P_o))$; the first and second terms represent the expected information gain and expected reward, respectively; and $c_t$ denotes the intensity of curiosity at time $t$. The heuristic idea of introducing the curiosity meta-parameter $c_t$ was also proposed in the RL field[30].

We briefly show how to derive it based on ref. 41. The free energy at current time $t$ is described by

$$F(o_t, \varphi_t) = E_{Q(\mathbf{w}_t|\varphi_t)}\left[\ln Q(\mathbf{w}_t|\varphi_t) - \ln P(o_t, \mathbf{w}_t|o_{1:t-1}, \mathbf{a}_{1:t})\right], \tag{31}$$

which is a rewriting of equation (17). Here, we attempt to express the future free energy at time $t + 1$, conditioned on action $\mathbf{a}_{t+1}$ as

$$F(o_{t+1}, \mathbf{a}_{t+1}) = E_{Q(\mathbf{w}_{t+1}|\varphi_t)}\left[\ln Q(\mathbf{w}_{t+1}|\varphi_t) - \ln P(o_{t+1}, \mathbf{w}_{t+1}|o_{1:t}, \mathbf{a}_{1:t+1})\right]. \tag{32}$$

However, there is an apparent problem in this equation: $o_{t+1}$ has not yet been observed because it is a future observation. To resolve this issue, we take an expectation with respect to $o_{t+1}$ as

$$F(\mathbf{a}_{t+1}) = E_{Q(o_{t+1}|\mathbf{w}_{t+1}, \mathbf{a}_{t+1})Q(\mathbf{w}_{t+1}|\varphi_t)}$$
$$\left[\ln Q(\mathbf{w}_{t+1}|\varphi_t) - \ln P(o_{t+1}, \mathbf{w}_{t+1}|o_{1:t}, \mathbf{a}_{1:t+1})\right]. \tag{33}$$

This can be rewritten as

$$F(\mathbf{a}_{t+1}) = E_{Q(o_{t+1}, \mathbf{w}_{t+1}|o_{1:t}, \mathbf{a}_{t+1})}\left[\ln Q(\mathbf{w}_{t+1}|\varphi_t)\right.$$
$$\left. - \ln P(\mathbf{w}_{t+1}|o_{1:t}, \mathbf{a}_{1:t+1}) - \ln P(o_{t+1}|o_{1:t}, \mathbf{a}_{1:t+1})\right] \tag{34}$$

Although $P(o_{t+1}|o_{1:t}, \mathbf{a}_{1:t+1})$ can be computed by the generative model as

$$P(o_{t+1}|o_{1:t}, \mathbf{a}_{1:t+1}) = \int P(o_{t+1}|\mathbf{w}_{t+1}, \mathbf{a}_{1:t+1})P(\mathbf{w}_{t+1}|o_{1:t})\,\mathrm{d}\mathbf{w}_{t+1}$$
$$\cong \int P(o_{t+1}|\mathbf{w}_{t+1}, \mathbf{a}_{1:t+1})Q(\mathbf{w}_{t+1}|\varphi_t)\,\mathrm{d}\mathbf{w}_{t+1}, \tag{35}$$

it is assumed that $P(o_{t+1}|o_{1:t}, \mathbf{a}_{1:t+1})$ was heuristically replaced with $P(o_{t+1})$ as the prior agent's preference of observing $o_{t+1}$, so that $\ln P(o_{t+1})$ can be addressed as an instrumental reward. The equation (34) was further transformed into

$$F(\mathbf{a}_{t+1}) = E_{Q(o_{t+1}|\mathbf{a}_{t+1})Q(\mathbf{w}_{t+1}|o_{1:t+1}, \mathbf{a}_{t+1})}\left[\ln Q(\mathbf{w}_{t+1}|\varphi_t)\right.$$
$$\left. - \ln Q(\mathbf{w}_{t+1}|o_{1:t+1}, \mathbf{a}_{1:t+1}) - \ln P(o_{t+1})\right]. \tag{36}$$

Finally, we obtained the so-called expected free energy as

$$F(\mathbf{a}_{t+1}) = E_{Q(o_{t+1}|\mathbf{a}_{t+1})}\left[-\text{KL}\left[Q(\mathbf{w}_{t+1}|o_{t+1}, \mathbf{a}_{t+1})\|Q(\mathbf{w}_{t+1})\right] - \ln P(o_{t+1})\right]. \tag{37}$$

where the KL divergence is called a Bayesian surprise, representing the extent to which the agent's beliefs are updated by observation, whereas the second term $\ln P(o_{t+1})$ can be addressed as the agent's prior preference of observing $o_{t+1}$, which can be interpreted as an instrumental reward. Therefore, the expected free energy is derived in an ad hoc manner.

In the first term of equation (30), the posterior and prior distributions of $\mathbf{w}_{t+1}$ in the KL divergence are derived as

$$Q(\mathbf{w}_{t+1}) = \int P(\mathbf{w}_{t+1}|\mathbf{w}_t)Q(\mathbf{w}_t)\,\mathrm{d}\mathbf{w}_t, \tag{38}$$

$$Q(\mathbf{w}_{t+1}|o_{t+1}, \mathbf{a}_{t+1}) = \frac{P(o_{t+1}|\mathbf{w}_{t+1}, \mathbf{a}_{t+1})Q(\mathbf{w}_{t+1}|\mathbf{a}_{t+1})}{P(o_{t+1}|\mathbf{a}_{t+1})}, \tag{39}$$

respectively. $o_{t+1}$ is a future observation, and therefore, the first term was expected by $P(o_{t+1}|\mathbf{a}_{t+1})$, which can be calculated as

$$P(o_{t+1}|\mathbf{a}_{t+1}) = \int P(o_{t+1}|\mathbf{w}_{t+1}, \mathbf{a}_{t+1})Q(\mathbf{w}_{t+1}|\mathbf{a}_{t+1})\,\mathrm{d}\mathbf{w}_{t+1}. \tag{40}$$

In the second term, the reward is quantitatively interpreted as the desired probability of $o_{t+1}$. For the two-choice task, we use

$$P(o_{t+1}) = P_o^{o_{t+1}}(1 - P_o)^{(1 - o_{t+1})}, \tag{41}$$

where $P_o$ indicates the desired probability of the presence of a reward. According to the probabilistic interpretation of reward[22,23],

the presence and absence of a reward can be evaluated by $\ln P_o$ and $\ln(1 - P_o)$, respectively.

In this study, we changed the sign of the expected free energy as the expected net utility, because we modeled decision-making as the maximization of the expected free energy. We ventured to introduce the curiosity meta-parameter $c_t$ to express irrational decision-making. Because rewards are relative, in this study, we set $\ln\{P_o/(1-P_o)\}$ and 0 for the presence and absence of a reward, respectively. Thus, our expected net utility can be described by equation (30), in which there is a gap between the original expected free energy and our expected net utility.

### Model for action selection

The agent probabilistically selects a choice with higher expected net utility as

$$P(\mathbf{a}_{t+1}) = \frac{\exp(\beta U(\mathbf{a}_{t+1}))}{\sum_{\mathbf{a}} \exp(\beta U(\mathbf{a}))}, \tag{42}$$

where $U(\mathbf{a}_{t+1})$ indicates the expected net utility of action $\mathbf{a}_{t+1}$. Equation (42) is equivalent to equation (2). To derive equation (42), we considered the expectation of the expected net utility with respect to probabilistic action as

$$E[U] = E_{Q(\mathbf{a}_{t+1})}\left[U(\mathbf{a}_{t+1}) - \beta^{-1}\ln Q(\mathbf{a}_{t+1})\right], \tag{43}$$

where $\beta$ indicates an inverse temperature, and the entropic constraint of action probability is introduced in the second term. This equation can be rewritten as

$$E[U] = -\beta^{-1}\text{KL}\left[Q(\mathbf{a}_{t+1}) \| \exp\left(\beta U(\mathbf{a}_{t+1})\right)/Z\right] + \beta^{-1}\ln Z, \tag{44}$$

where $Z$ indicates a normalization constant. Thus, its maximization respective to $Q(\mathbf{a}_{t+1})$ leads to the optimal action probability as shown in equation (42).

### Alternative expected net utility

For comparison, we consider an alternative expected net utility by introducing a time-dependent meta-parameter in the second term as follows:

$$U(\mathbf{a}_{t+1}) = E_{P(o_{t+1}|\mathbf{a}_{t+1})}\left[\text{KL}\left[Q(\mathbf{w}_{t+1}|o_{t+1},\mathbf{a}_{t+1})\|Q(\mathbf{w}_{t+1}|\mathbf{a}_{t+1})\right]\right] \\ + d_t \cdot E_{P(o_{t+1}|\mathbf{a}_{t+1})}\left[R(o_{t+1})\right], \tag{45}$$

where $d_t$ denotes the subjective intensity of reward at time $t$. In this case, the agent with high $d_t$ will show more exploitative behavior, whereas the agent with $d_t = 0$ shows more explorative behavior driven by the expected information gain.

### Calculation of expected net utility

Here, we present the calculation of the expected net utility. The KL divergence in the first term of equation (30) can be transformed into

$$E_{P(o_{t+1}|\mathbf{a}_{t+1})}\left[\text{KL}\left[Q(\mathbf{w}_{t+1}|o_{t+1},\mathbf{a}_{t+1})\|Q(\mathbf{w}_{t+1}|\mathbf{a}_{t+1})\right]\right] = H(o_{t+1}) - H(o_{t+1}|\mathbf{w}_{t+1}), \tag{46}$$

where the first and second terms represent the conditional and marginal entropies, respectively:

$$H(o_{t+1}|\mathbf{w}_{t+1}) = E_{P(o_{t+1}|\mathbf{w}_{t+1},\mathbf{a}_{t+1})Q(\mathbf{w}_{t+1}|\mathbf{a}_{t+1})}\left[-\ln P(o_{t+1}|\mathbf{w}_{t+1},\mathbf{a}_{t+1})\right], \tag{47}$$

$$H(o_{t+1}) = E_{P(o_{t+1}|\mathbf{a}_{t+1})}\left[-\ln P(o_{t+1}|\mathbf{a}_{t+1})\right]. \tag{48}$$

The conditional entropy $H(o_{t+1}|\mathbf{w}_{t+1})$ can be calculated by substituting equation (13) into equation (47) as

$$H(o_{t+1}|\mathbf{w}_{t+1}) = -E_{Q(\mathbf{w}_{t+1}|\mathbf{a}_{t+1})}\left[\sum_i a_{i,t+1}g(w_{i,t+1})\right], \tag{49}$$

where

$$g(w) = f(w)\ln f(w) + (1 - f(w))\ln(1 - f(w)). \tag{50}$$

Here, we approximately calculate this equation by using the second-order Taylor expansion as

$$H(o_{t+1}|\mathbf{w}_{t+1}) \cong -E_{Q(\mathbf{w}_{t+1}|\mathbf{a}_{t+1})}\left[\sum_i a_{i,t+1}\left\{\begin{array}{c}g(\mu_{i,t+1}) + \frac{\partial g}{\partial w_{i,t+1}}(w_{i,t+1} - \mu_{i,t+1}) \\ + \frac{1}{2}\frac{\partial^2 g}{\partial w_{i,t+1}^2}(w_{i,t+1} - \mu_{i,t+1})^2\end{array}\right\}\right], \tag{51}$$

which leads to

$$H(o_{t+1}|\mathbf{w}_{t+1}) = \\ -\sum_i a_{i,t+1}\left[\begin{array}{c}f(\mu_{i,t+1})\ln f(\mu_{i,t+1}) + (1 - f(\mu_{i,t+1}))\ln(1 - f(\mu_{i,t+1})) \\ + \frac{1}{2}\left\{f(\mu_{i,t+1})(1 - f(\mu_{i,t+1}))\left(1 + (1 - 2f(\mu_{i,t+1}))\ln\frac{f(\mu_{i,t+1})}{1 - f(\mu_{i,t+1})}\right)\right\} \\ (p_{i,t}^{-1} + p_w^{-1})\end{array}\right]. \tag{52}$$

The marginal entropy $H(o_{t+1})$ can be calculated as

$$H(o_{t+1}) = -\sum_i a_{i,t+1}\{P(o_{t+1} = 0|\mathbf{a}_{t+1})\ln P(o_{t+1} = 0|\mathbf{a}_{t+1}), \\ + P(o_{t+1} = 1|\mathbf{a}_{t+1})\ln P(o_{t+1} = 1|\mathbf{a}_{t+1})\} \tag{53}$$

where

$$P(o_{t+1}|\mathbf{a}_{t+1}) = \int P(o_{t+1}|\mathbf{w}_{t+1},\mathbf{a}_{t+1})Q(\mathbf{w}_{t+1}|\mathbf{a}_{t+1})\,d\mathbf{w}_{t+1}$$

$$= \int \prod_i \left\{f(w_{i,t+1})^{o_{t+1}}(1 - f(w_{i,t+1}))^{1-o_{t+1}}\right\}^{a_{i,t+1}} Q(\mathbf{w}_{t+1}|\mathbf{a}_{t+1})\,d\mathbf{w}_{t+1}$$

$$= \prod_i \left[\begin{array}{c}f(\mu_{i,t+1})^{o_{t+1}}(1 - f(\mu_{i,t+1}))^{1-o_{t+1}} \\ + 1^{o_{t+1}}(-1)^{1-o_{t+1}}\frac{1}{2}f(\mu_{i,t+1})\{1 - f(\mu_{i,t+1})\}\{1 - 2f(\mu_{i,t+1})\}(p_{i,t}^{-1} + p_w^{-1})\end{array}\right]^{a_{i,t+1}}. \tag{54}$$

The second term of the expected net utility (equation (36)) is calculated as

$$E_{P(o_{t+1}|\mathbf{a}_{t+1})}\left[\ln P(o_{t+1})\right] = E_{P(o_{t+1}|\mathbf{a}_{t+1})}\left[o_{t+1}\ln P_o + (1 - o_{t+1})\ln(1 - P_o)\right] \tag{55}$$

$$= P(o_{t+1} = 0|\mathbf{a}_{t+1})\ln(1 - P_o) + P(o_{t+1} = 1|\mathbf{a}_{t+1})\ln(1 - P_o).$$

### Observer-SSM

We constructed the observer-SSM, which describes the temporal transitions of the latent internal state z of agent and the generation of action, from the viewpoint of the observer of the agent. This is depicted graphically in Fig. 4. As prior information, we assumed that the agent acts based on the internal state, that is, the intensity of curiosity, the recognized reward probabilities and their confidence levels. The intensity of curiosity was assumed to change temporally as a random walk:

$$c_t = c_{t-1} + \epsilon\zeta_t, \tag{56}$$

where $\zeta_t$ denotes white noise with zero mean and unit variance, and $\epsilon$ denotes its noise intensity. Other internal states, that is, $\mu_i$ and $p_i$, were assumed to update as equations (28) and (29). The transition of the internal state is expressed by the probability distribution

$$P(\mathbf{z}_t|\mathbf{z}_{t-1}) = \mathcal{N}(\mathbf{z}_t|\mathbf{F}(\mathbf{z}_{t-1}, \mathbf{a}_{t-1}), \Gamma), \tag{57}$$

$$\mathbf{F}(\mathbf{z}_{t-1}, \mathbf{a}_{t-1}) = \begin{bmatrix} 0 \\ h(\mu_{1,t-1}, p_{1,t-1}, o_t, a_1) \\ h(\mu_{2,t-1}, p_{2,t-1}, o_t, a_2) \\ k(\mu_{1,t-1}, p_{1,t-1}, a_1) \\ k(\mu_{2,t-1}, p_{2,t-1}, a_2) \end{bmatrix}, \tag{58}$$

where $\mathbf{z}_t = (c_t, \boldsymbol{\mu}_t^\mathsf{T}, \mathbf{p}_t^\mathsf{T})^\mathsf{T}$ and $\Gamma = \epsilon^2 \mathrm{diag}(1, 0, 0, 0, 0)$. $h(\mu_{i,t-1}, p_{i,t-1}, o_t, a_i)$ and $k(\mu_{i,t-1}, p_{i,t-1}, a_i)$ represent the right-hand sides of equations (28) and (29), respectively; and $\Gamma$ and diag $(\mathbf{x})$ denote the variance–covariance matrix and square matrix whose diagonal component is $\mathbf{x}$, respectively. In addition, the agent was assumed to select an action $\mathbf{a}_{t+1}$ based on the expected net utilities, as follows:

$$P(\mathbf{a}_{t+1}) = \frac{\exp(\beta U(\mathbf{a}_{t+1}))}{\sum_{\mathbf{a}} \exp(\beta U(\mathbf{a}))}, \tag{59}$$

and the reward was obtained by the following probability distribution:

$$P(o_t|\mathbf{a}_t) = \prod_i \left\{ \lambda_{i,t}^{o_t}(1 - \lambda_{i,t})^{1-o_t} \right\}^{a_{i,t}}. \tag{60}$$

## Q-leaning in two-choice task and its observer-SSM

The decision-making in the two-choice task was also modeled by Q-learning. Reward prediction for the $i$th option $Q_{i,t}$ is updated as

$$Q_{i,t} = Q_{i,t-1} + \alpha_{t-1}(r_t a_{i,t-1} - Q_{i,t-1}), \tag{61}$$

where $\alpha_t$ indicates a learning rate at trail $t$. The agents selected action following a softmax function:

$$P(a_{i,t} = 1) = \frac{\exp(B_t Q_{i,t})}{\sum_i \exp(B_t Q_{i,t})}, \tag{62}$$

where $B_t$ indicates the inverse temperature at trail $t$ controlling the randomness of the action selection.

The time-dependent parameters $\alpha_t$ and $B_t$ in Q-learning were estimated from behavioral data[37]. These parameters were assumed to change temporally as a random walk:

$$\theta_t = \theta_{t-1} + \epsilon_\theta \zeta_{\theta,t}, \tag{63}$$

where $\theta \in \{\alpha, B\}$, $\zeta_{\theta,t}$ denotes white noise with zero mean and unit variance and $\epsilon_\theta$ denotes its noise intensity. Thus, the transition of the internal state is expressed by the probability distribution

$$P(\mathbf{z}_t|\mathbf{z}_{t-1}) = \mathcal{N}(\mathbf{z}_t|\mathbf{F}(\mathbf{z}_{t-1}, \mathbf{a}_{t-1}), \Gamma), \tag{64}$$

$$\mathbf{F}(\mathbf{z}_{t-1}, \mathbf{a}_{t-1}) = \begin{bmatrix} 0 \\ 0 \\ h_1(\alpha_t, Q_{1,t-1}, \mathbf{a}_{t-1}) \\ h_2(\alpha_t, Q_{2,t-1}, \mathbf{a}_{t-1}) \end{bmatrix}, \tag{65}$$

where $\mathbf{z}_t = (\alpha_t, B_t, Q_{1,t}, Q_{2,t})^\mathsf{T}$, $\Gamma = \epsilon^2 \mathrm{diag}(1, 1, 0, 0)$, $h_i(\alpha_t, Q_{i,t-1}, \mathbf{a}_t)$ represents the right-hand side of equation (61); and $\Gamma$ and diag $(\mathbf{x})$ denote the variance–covariance matrix and square matrix whose diagonal component is $\mathbf{x}$, respectively.

## iFEP by particle filter and Kalman backward algorithm

Based on the observer-SSM, we estimated the posterior distribution of the latent internal state of agent $z_t$ given all observations from 1 to $T(x_{1:T})$ in a Bayesian manner, that is, $P(z_t|x_{1:T})$. This estimation was done by forward and backward algorithms, which are called filtering and smoothing, respectively.

In filtering, the posterior distribution of $z_t$ given observations until $t(x_{1:t})$ is sequentially updated in a forward direction as

$$P(\mathbf{z}_t|\mathbf{x}_{1:t}) \propto P(\mathbf{x}_t|\mathbf{z}_t, \theta) \int P(\mathbf{z}_t|\mathbf{z}_{t-1}, \theta) P(\mathbf{z}_{t-1}|\mathbf{x}_{1:t-1}) \, \mathrm{d}\mathbf{z}_{t-1}, \tag{66}$$

where $\mathbf{x}_t = (\mathbf{a}_t^T, o_t)^T$ and $\theta = \{\sigma^2, \alpha, P_o\}$. The prior distribution of $\mathbf{z}_1$ is

$$P(\mathbf{z}_1) = \left[ \prod_i \mathcal{N}(\mu_{i,1}|\mu_0, \sigma_\mu^2) \mathrm{Gam}(p_{i,1}|a_g, b_g) \right] \mathrm{Uni}(c_1|a_u, b_u), \tag{67}$$

where $\mu_0$ and $\sigma_\mu^2$ denote means and variances, Gam$(x|a_g, b_g)$ indicates the Gamma distribution with shape parameter $a_g$ and scale parameter $b_g$, and Uni$(x|a_u, b_u)$ indicates uniform distribution from $a_u$ to $b_u$. We used a particle filter[42] to sequentially calculate the posterior $P(\mathbf{z}_t|\mathbf{x}_{1:t})$, which cannot be analytically derived because of the nonlinear transition probability.

After the particle filter, the posterior distribution of $\mathbf{z}_t$ given all observations $(\mathbf{x}_{1:T})$ is sequentially updated in a backward direction as

$$P(\mathbf{z}_t|\mathbf{x}_{1:T}) = \int P(\mathbf{z}_{t+1}|\mathbf{x}_{1:T}) P(\mathbf{z}_t|\mathbf{z}_{t+1}, \mathbf{x}_{1:t}, \theta) \, \mathrm{d}\mathbf{z}_{t+1} \tag{68}$$

$$= \int P(\mathbf{z}_{t+1}|\mathbf{x}_{1:T}) \frac{P(\mathbf{z}_{t+1}|\mathbf{z}_t, \theta) P(\mathbf{z}_t|\mathbf{x}_{1:t}, \theta)}{\int P(\mathbf{z}_{t+1}|\mathbf{z}_t, \theta) P(\mathbf{z}_t|\mathbf{x}_{1:t}, \theta) \mathrm{d}\mathbf{z}_t} \, \mathrm{d}\mathbf{z}_{t+1}.$$

However, this backward integration is intractable because of the non-Gaussian $P(\mathbf{z}_T|\mathbf{x}_{1:T})$, which was represented by the particle ensemble in the particle filter, and the nonlinear relationship between $\mathbf{z}_t$ and $\mathbf{z}_{t+1}$ in $P(\mathbf{z}_{t+1}|\mathbf{z}_t, \theta)$ (equation (57)). Thus, we approximated $P(\mathbf{z}_t|\mathbf{x}_{1:t})$ as $\mathcal{N}(\mathbf{z}_t|\mathbf{m}_t, \mathbf{V}_t)$, where $\mathbf{m}_t$ and $\mathbf{V}_t$ denote a sample mean and a sample variance of the particles at $t$, whereas we linearized $P(\mathbf{z}_t|\mathbf{z}_{t-1}, \theta)$ as

$$P(\mathbf{z}_t|\mathbf{z}_{t-1}, \theta) \cong \mathcal{N}(\mathbf{z}_t|\mathbf{A}\mathbf{z}_t + \mathbf{b}, \Gamma), \tag{69}$$

$$\mathbf{A} = \left. \frac{\partial \mathbf{F}(\mathbf{z}_{t-1}, \mathbf{a}_{t-1})}{\partial \mathbf{z}_{t-1}} \right|_{\mathbf{m}_t}, \tag{70}$$

$$\mathbf{b} = F(\mathbf{m}_t, \mathbf{a}_{t-1}) - A\mathbf{m}_t, \tag{71}$$

where $\mathbf{A}$ denotes a Jacobian matrix. Because these approximations make the integration of equation (68) tractable, the posterior distribution $P(\mathbf{z}_t|\mathbf{x}_{1:T})$ can be computed by a Gaussian distribution as

$$P(\mathbf{z}_t|\mathbf{x}_{1:T}) = \mathcal{N}(\mathbf{z}_t|\widehat{\mathbf{m}}_t, \widehat{\mathbf{V}}_t) \tag{72}$$

whose mean and variance were analytically updated by Kalman backward algorithms as[43]

$$\widehat{\mathbf{m}}_t = \mathbf{m}_t + \mathbf{J}_t \{\widehat{\mathbf{m}}_{t+1} - (\mathbf{A}\mathbf{m}_t + \mathbf{b})\}, \tag{73}$$

$$\widehat{\mathbf{V}}_t = \mathbf{V}_t + \mathbf{J}_t \{\widehat{\mathbf{m}}_{t+1} - (\mathbf{A}\mathbf{V}_t\mathbf{A}^\mathsf{T} + \Gamma)\}\mathbf{J}_t^\mathsf{T}, \tag{74}$$

where

$$\mathbf{J}_t = \mathbf{V}_t\mathbf{A}^\mathsf{T}(\mathbf{A}\mathbf{V}_t\mathbf{A}^\mathsf{T} + \Gamma)^{-1}. \tag{75}$$

## Impossibility of model discrimination

In the ReCU and Q-learning models, the action selections were formulated with the same softmax functions as

$$P(a_{i,t} = 1) = \frac{\exp(\beta(E[\text{Reward}_{i,t}] + c_t E[\text{Info}_{i,t}]))}{\sum_j \exp(\beta(E[\text{Reward}_{j,t}] + c_t E[\text{Info}_{j,t}]))}. \tag{76}$$

$$P(a_{i,t} = 1) = \frac{\exp(\beta_t Q_{i,t})}{\sum_i \exp(\beta_t Q_{i,t})}, \tag{77}$$

which correspond to that of the ReCU and Q-learning models, respectively. These equations contain the time-dependent meta-parameters, that is, $c_t$ and $\beta_t$. For both models, the goodness of fit (that is, likelihood) for the actual behavioral data can be freely improved by tuning the time-dependent meta-parameters. Thus, discrimination between the ReCU and Q-learning models must be essentially impossible.

## Estimation of parameters in iFEP

The ReCU model has several parameters: $\sigma_w^2, \alpha, \beta, P_o$ and $\epsilon$. In the estimation, we set $\epsilon$ to 1, which was the optimal value for estimation in the artificial data (Supplementary Fig. 3). We assumed the unit intensity of reward, that is, $\ln P_o/(1 - P_o) = 1$, because it is impossible to estimate both $P_o$ and $\beta$ caused by multiplying $\beta$ and $\ln P_o/(1 - P_o)$ in the expected net utility (equations (30) and (42)). This treatment is suitable for relative comparison between the curiosity meta-parameter and the reward. In addition, we addressed $\beta c_t$ as a latent variable as $\hat{c}_t = \beta c_t$ because of the multiplication of $\beta$ in the expected net utility (equations (30) and (42)). Thus, the estimation of $c_t$ can be obtained by dividing the estimated $\hat{c}_t$ by the estimated $\beta$. Therefore, the hyperparameters to be estimated were $\sigma_w^2, \alpha$ and $\beta$.

To estimate these parameters $\theta = \{\sigma_w^2, \alpha, \beta\}$, we extended the observer-SSM to a self-organizing SSM[44] in which $\theta$ was addressed as constant latent variables:

$$P(\mathbf{z}_t, \theta | \mathbf{x}_{1:t}) \propto P(\mathbf{x}_t | \mathbf{z}_t) \int P(\mathbf{z}_t | \mathbf{z}_{t-1}, \theta) P(\mathbf{z}_{t-1}, \theta | \mathbf{x}_{1:t-1}) \, d\mathbf{z}_{t-1}, \tag{78}$$

where $P(\theta) = \text{Uni}(\sigma^2 | a_\sigma, b_\sigma) \text{Uni}(\alpha | a_\alpha, b_\alpha) \mathcal{N}(\beta | m_\beta, v_\beta)$. To sequentially calculate the posterior $P(\mathbf{z}_t, \theta | \mathbf{x}_{1:t})$ using the particle filter, we used 100,000 particles and augmented the state vector of all particles by adding the parameter $\theta$, which was not updated from randomly sampled initial values.

The hyperparameter values used in this estimation were $\mu_0 = 0$, $\sigma_\mu^2 = 0.01^2, a_g = 10, b_g = 0.001, a_u = -15, b_u = 15, a_\sigma = 0.2, b_\sigma = 0.7, a_\alpha = 0.04, b_\alpha = 0.06, a_\beta = 0$ and $b_\beta = 50$, which were heuristically given as parameters correctly estimated using the artificial data (Supplementary Fig. 2).

## Statistical testing with Monte Carlo simulations

Supplementary Fig. 5 shows statistical testing of the negative curiosity estimated in Fig. 5. A null hypothesis is that an agent has no curiosity (that is, $c_t = 0$) decides on a choice only depending on its recognition of the reward probability. Under the null hypothesis, model simulations were repeated 1,000 times under the same experimental conditions as in Fig. 5 and the curiosity was estimated for each using iFEP. We adopted the temporal average of the estimated curiosity as a test statistic and plotted the null distribution of the test statistic. Compared with the estimated curiosity of the rat behavior, we computed the $P$ value for a one-sided left-tailed test.

## Reporting summary

Further information on research design is available in the Nature Portfolio Reporting Summary linked to this article.

## Data availability

Source data for Figs. 2, 3, 5 and 6 are available with this paper. Source data for Supplementary Figures are available in Supplementary Data.

We used the rat behavioral data published in ref. 36, which is publicly available at https://groups.oist.jp/ja/ncu/data. These rat behavioral data are also included in the Source Data for Fig. 5 and on Zenodo[45].

## Code availability

The computer simulation and data analysis were performed using MATLAB (version R2020b) software. The code used for this work is available on GitHub at https://github.com/YukiKonaka/Konaka_Honda_2023. The specific version used to produce the results in this manuscript is also available on Zenodo[45].

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

## Acknowledgements

We are grateful to K. Doya and M. Ito for providing rat behavioral data. We thank the organizers of the tutorial on the free energy principle in 2019, which inspired this research, and I. Higashino and M. Fujiwara-Yada for carefully checking all the equations in the manuscript. This study was supported in part by a Grant-in-Aid for Transformative Research Areas (B) (no. 21H05170), AMED (grant no. JP21wm0425010), Moonshot R&D–MILLENNIA program (grant no. JPMJMS2024-9) by JST, the Cooperative Study Program of Exploratory Research Center on Life and Living Systems (ExCELLS) (program no. 21-102) and the grant of Joint Research by the National Institutes of Natural Sciences (NINS program no. 01112102).

## Author contributions

H.N. conceived of the project. Y. K. and H.N. developed the method, and Y.K. implemented the model simulation. Y.K. and H.N. wrote the manuscript.

## Competing interests

The authors declare no competing interests.

## Additional information

**Correspondence and requests for materials** should be addressed to Honda Naoki.

# Reporting Summary

Nature Research wishes to improve the reproducibility of the work that we publish. This form provides structure for consistency and transparency in reporting. For further information on Nature Research policies, see our Editorial Policies and the Editorial Policy Checklist.

## Statistics

For all statistical analyses, confirm that the following items are present in the figure legend, table legend, main text, or Methods section.

| n/a | Confirmed | |
|---|---|---|
| ☐ | ☒ | The exact sample size ($n$) for each experimental group/condition, given as a discrete number and unit of measurement |
| ☒ | ☐ | A statement on whether measurements were taken from distinct samples or whether the same sample was measured repeatedly |
| ☐ | ☒ | The statistical test(s) used AND whether they are one- or two-sided<br>*Only common tests should be described solely by name; describe more complex techniques in the Methods section.* |
| ☒ | ☐ | A description of all covariates tested |
| ☐ | ☒ | A description of any assumptions or corrections, such as tests of normality and adjustment for multiple comparisons |
| ☐ | ☒ | A full description of the statistical parameters including central tendency (e.g. means) or other basic estimates (e.g. regression coefficient) AND variation (e.g. standard deviation) or associated estimates of uncertainty (e.g. confidence intervals) |
| ☐ | ☒ | For null hypothesis testing, the test statistic (e.g. $F$, $t$, $r$) with confidence intervals, effect sizes, degrees of freedom and $P$ value noted<br>*Give P values as exact values whenever suitable.* |
| ☐ | ☒ | For Bayesian analysis, information on the choice of priors and Markov chain Monte Carlo settings |
| ☒ | ☐ | For hierarchical and complex designs, identification of the appropriate level for tests and full reporting of outcomes |
| ☐ | ☒ | Estimates of effect sizes (e.g. Cohen's $d$, Pearson's $r$), indicating how they were calculated |

*Our web collection on statistics for biologists contains articles on many of the points above.*

## Software and code

Policy information about availability of computer code

| Data collection | No software was used for data collection |
|---|---|
| Data analysis | The computer simulation and data analysis were done using Matlab software (Version R2020b). The code used for this work are available on GitHub at: https://github.com/YukiKonaka/Konaka_Honda_2023. The specific version used to produce the results in this manuscript is also available on Zenodo at https://doi.org/10.5281/zenodo.7722905.<br><br>The algorithm used in this study is particle filter in control system toolbox (Version R2020b). We also used shaded ErrorBar (GNU LESSER GENERAL PUBLIC LICENSE Version 3, 29 June 2007), which is public MATLAB function located at https://github.com/raacampbell/shadedErrorBar/blob/master/shadedErrorBar.m. The file of shaded ErrorBar was also uploaded at our GitHub (https://github.com/YukiKonaka/Konaka_Honda_2023) and our Zenodo (https://doi.org/10.5281/zenodo.7722905). |

For manuscripts utilizing custom algorithms or software that are central to the research but not yet described in published literature, software must be made available to editors and reviewers. We strongly encourage code deposition in a community repository (e.g. GitHub). See the Nature Research guidelines for submitting code & software for further information.

## Data

Policy information about availability of data

All manuscripts must include a data availability statement. This statement should provide the following information, where applicable:
- Accession codes, unique identifiers, or web links for publicly available datasets
- A list of figures that have associated raw data
- A description of any restrictions on data availability

Source data for figures 2, 3, 5 and 6 are available with this paper. Source data for Supplementary Figures are available in Supplementary Data. We used the rat

behavioral data published by Makoto Ito and Kenji Doya, Validation of Decision-Making Models and Analysis of Decision Variables in the Rat Basal Ganglia, Journal of Neuroscience (Publisher: the Society for Neuroscience), 2009 (DOI: 10.1523/JNEUROSCI.6157-08.2009), which is publicly available at Prof. Doya Kenji's homepage: https://groups.oist.jp/ja/ncu/data. This rat behavioral data is also included in Source data for figure 5 and Zenodo at https://doi.org/10.5281/zenodo.7722905.

# Field-specific reporting

Please select the one below that is the best fit for your research. If you are not sure, read the appropriate sections before making your selection.

☒ Life sciences ☐ Behavioural & social sciences ☐ Ecological, evolutionary & environmental sciences

For a reference copy of the document with all sections, see nature.com/documents/nr-reporting-summary-flat.pdf

# Life sciences study design

All studies must disclose on these points even when the disclosure is negative.

| | |
|---|---|
| Sample size | Data size is based on the rat behavioral data from Ito & Doya Journal of Neuroscience 2009, which was cited in the manuscript. |
| Data exclusions | In the rat data of the two-choice task (Ito & Doya Journal of Neuroscience 2009), a rat selected left or right, but sometimes fail to select. In our analysis, we excluded no-choice trials from the behavioral time-series because we assumed the rat cannot update the recognition because of no observation of reward in the no-choice trials. |
| Replication | We checked the estimation performance by replicating the estimations (Supplementary Fig. 3). |
| Randomization | This is not relevant to our study because we only decipher the temporal dynamics of the internal state including curiosity meta-parameter from the datasets. There is not allocation procedure. |
| Blinding | No data collection was involved in the present study. Blinding was not possible because we only decode curiosity from the datasets, there is no allocation procedure. |

# Reporting for specific materials, systems and methods

We require information from authors about some types of materials, experimental systems and methods used in many studies. Here, indicate whether each material, system or method listed is relevant to your study. If you are not sure if a list item applies to your research, read the appropriate section before selecting a response.

## Materials & experimental systems

| n/a | Involved in the study |
|---|---|
| ☒ | ☐ Antibodies |
| ☒ | ☐ Eukaryotic cell lines |
| ☒ | ☐ Palaeontology and archaeology |
| ☒ | ☐ Animals and other organisms |
| ☒ | ☐ Human research participants |
| ☒ | ☐ Clinical data |
| ☒ | ☐ Dual use research of concern |

## Methods

| n/a | Involved in the study |
|---|---|
| ☒ | ☐ ChIP-seq |
| ☒ | ☐ Flow cytometry |
| ☒ | ☐ MRI-based neuroimaging |

