## [Peer Review File · Nature Computational Science]

Peer Review Information

Journal: Nature Computational Science

Manuscript Title: Decoding reward–curiosity conflict in decision-making from irrational behaviors

Corresponding author name(s): Honda Naoki

Reviewer Comments & Decisions:

Decision Letter, initial version:
--

Date: 3rd October 22 16:16:04

Last Sent: 3rd October 22 16:16:04

Triggered By: Ananya Rastogi

From: ananya.rastogi@nature.com

To: nhonda@hiroshima-u.ac.jp

BCC: ananya.rastogi@nature.com

Subject: Decision on Nature Computational Science manuscript NATCOMPUTSCI-22-0624

Message: ** Please ensure you delete the link to your author homepage in this e-mail if you wish to forward it to your co-authors. **

Dear Dr Naoki,

Your manuscript "Decoding reward–curiosity conflict in decision-making from irrational behaviors" has now been seen by 2 referees, whose comments are appended below. You will see that while they find your work of interest, they have raised points that need to be addressed before we can make a decision on publication.

The referees' reports seem to be quite clear. Naturally, we will need you to address all of the points raised.

While we ask you to address all of the points raised, the following points need to be substantially worked on:

- Reviewer #1 has mentioned that due to the focus on time sensitive curiosity, you may have accidentally missed the important dynamic ranges in which reward and curiosity trade off against each other. To counter this, please introduce time

sensitivity by changing the precision of the reward preferences.

- Please carry out the analyses and iFEP but applying c to expected reward, where the log preferences for the presence or absence of an outcome are one natural unit or zero.
- Please fit the alpha parameter to improve interpretability of results.
- Please derive Equation 3 from the minimization process of the variational free energy.
- As requested by Reviewer #2, please show that the proposed iFEP model is significantly superior to existing models that can consider the dynamics of the meta-parameters such as (Samejima et al., 2005), in terms of explainability of observed behavioral data.
- Please discuss if iFEP has a higher generalization ability than existing models for behavioral data analysis.
- Please specify the hyperparameters values or show the basic policy to tune those values.

In addition to these points, it would also be beneficial to address the following concerns:

Please use the following link to submit your revised manuscript and a point-by-point response to the referees' comments (which should be in a separate document to any cover letter):

[REDACTED]

** This url links to your confidential homepage and associated information about manuscripts you may have submitted or be reviewing for us. If you wish to forward this e-mail to co-authors, please delete this link to your homepage first. **

To aid in the review process, we would appreciate it if you could also provide a copy of your manuscript files that indicates your revisions by making use of Track Changes or similar mark-up tools. Please also ensure that all correspondence is marked with your Nature Computational Science reference number in the subject line.

In addition, please make sure to upload a Word Document or LaTeX version of your text, to assist us in the editorial stage.

To improve transparency in authorship, we request that all authors identified as 'corresponding author' on published papers create and link their Open Researcher and Contributor Identifier (ORCID) with their account on the Manuscript Tracking System (MTS), prior to acceptance. ORCID helps the scientific community achieve unambiguous attribution of all scholarly contributions. You can create and link your ORCID from the home page of the MTS by clicking on 'Modify my Springer Nature account'. For more information please visit www.springernature.com/orcid.

We hope to receive your revised paper within three weeks. If you cannot send it within this time, please let us know.

Best regards,

Ananya Rastogi, PhD
Associate Editor
Nature Computational Science

Reviewers comments:

Reviewer #1 (Remarks to the Author):

I enjoyed reading this sophisticated numerical and empirical study of choice behaviour under the free energy principle. In terms of the ambition and potential importance of this work, I thought this was a substantial contribution. If you can develop an inverse FEP procedure to estimate belief updating, confidence and curiosity in animal studies, then this opens the door to potentially important applications in translational (e.g., pharmacological) studies. I was also impressed by your mathematical fluency in translating the free energy principle into your particular generative model – and the application of variational procedures (and particle filtering) to estimate hidden belief states of experimental subjects.

Although I think it is important that you bring this work to closure, there are some foundational issues you need to address first. I suspect that these will require a simplification of your model and a reanalysis of the empirical data. This is because your extension to the FEP is not licensed or necessary.

Because you have focused on time sensitive curiosity, as opposed to preferences, you have accidentally missed the important dynamic ranges in which reward and curiosity trade off against each other. Specifically, what you have studied is called “active inference” that can be derived from the free energy principle. The free energy principle is a first principle account based upon statistical physics. For your interest, you can read about where the expected free energy comes from in this recent summary:

<https://arxiv.org/ftp/arxiv/papers/2201/2201.06387.pdf>

The key point here is that the functional form of expected free energy (something you have called expected utility) cannot be changed. However, you can introduce a time sensitivity by changing the precision of the reward preferences. In other words, instead of applying c to the expected information gain, you can apply it to the reward preferences. This means that c now plays the role of a precision of preferences about outcomes. If you take this approach, you will be able to fit your empirical data more sensibly.

The problem with the current simulations — and the empirical estimates — is that c can be negative. However, the expected information gain is nonnegative. This means that you are explaining behaviour in terms of an aversion to expected information gain. This is a consequence of specifying reward preferences as a logit function (your first equation). The problem with the logit function is that the important variations in

preference are limited to an extremely small range; namely between $P_0 = 0.99 - 1$.

The key insight here is that expected information gain and expected reward in active inference have natural units (when using natural logarithms). This means a strong preference requires a difference in log probabilities of about 2 to 3. The logit function only produces these differences with values of P_0 that are close to one. Furthermore, the logic function places an upper bound on P_0 that is unnecessary with no precedent for this functional form.

Could you redo your analyses and iFEP but applying c to expected reward, where the log preferences for the presence or absence of an outcome are one natural unit or zero.

Expected free energy = Expected information gain + Expected reward

Expected reward = $E[\log p(o_{t+1})] = c \cdot E[\log p_0(o_{t+1})]$

Prior preferences = $p(o_{t+1} = \text{absent}) = 1 - P_0^c$
 $= p(o_{t+1} = \text{present}) = P_0^c$

Where $P_0 = 1$ is a baseline preference and c is a meta-parameter that sets the precision prior preferences.

You should then find an interesting switch between curiosity and reward with values of c that are around the expected information gain. I notice that your expected information gain was very small. This may be because you are using a continuous hidden state and a Gaussian or Laplace approximation to the posterior. It would be useful to start off with a prior over the hidden state w so that the entropy is nontrivial (e.g., 5 natural units greater than the posterior entropy after a typical number of trials (e.g., 32)).

What you should see is that as the simulated rat becomes more confident about the values of w , the expected information gain falls by a nontrivial amount. At present, it looks as if expected information gain varies by a trivial amount over several hundred trials, which means something is wrong with your setup. In other words, it looks as if the (synthetic) rats are not really learning anything about the payoff probabilities. In consequence, the differences in expected free energy between different actions may be very small and therefore it will be difficult to estimate the belief updating from observed choice behaviour.

A crucial point here, is that you have to decide whether action is sampled from the posterior beliefs about choices or whether you select the most likely action. Usually, when using active inference to fit choice behaviour in humans, one uses a further softmax precision or inverse temperature parameter (called α) that is applied to the expected free energy before applying the softmax operator. This allows you to fit a subject-specific stochasticity to action selection and obtain veridical estimates of the parameters that underwrite belief updating.

You appear not to have fitted this (α) parameter and therefore I think it is difficult to interpret your results; especially when allowing for a random walk on c . I do not think you need to model random walks or volatility on either the expected information

gain or the reward preferences. This is because active inference explains time-dependent changes in the balance between curiosity and reward. This follows because as exploration increases confidence about latent states (i.e., w) the expected information gain falls to 0 and curiosity gives way to exploitative, preference-satisfying behaviour.

This is a fundamental aspect of active inference that you would expect to see in your simulations and in the empirical data that you analyse. If the empirical data is driven largely by preferences with minimal exploration, then this should be explained sufficiently by having a large value of c applied to the baseline prior preferences above. In other words, adding time-dependent precisions to expected free energy is interesting but may make it difficult for you to invert the active inference (FEP) model without introducing lots of conditional independencies. I know that this is the case because there is quite a lot of experience of doing this in the field of active inference; particularly, in computational phenotyping in psychiatry. An early example can be found here [1]

Suggestions for the paper:

Please replace the sentences in the introduction starting with "Furthermore, FEP was.... between reward and curiosity." With something like:

"The free energy principle is a first principle account of sentient behaviour that is motivated from the statistical physics of self organisation. From the perspective of behavioural science and decision-making, the most interesting contribution of the FEP is an expected free energy that scores the likelihood of an action in the setting of active (planning as) inference [2-6]. The ensuing formulation of action and perception, in terms of active inference, accommodates reward learning by formulating rewards as preferences over outcomes. This allows one to express both the information-seeking and preference-seeking contributions to expected free energy in the same units (usually natural units when using natural logarithms). This rests upon the decomposition of expected free energy into expected information gain and expected reward. These two components underwrite optional Bayesian design [7, 8] and decision theoretic [9] formulations of behaviour respectively.

Crucially, the balance between the two components depends upon the precision of the prior preferences that underwrite subjective reward or utility. Agents with very precise prior preferences will engage in more exploitative behaviour; whereas — in the absence of prior preferences — explorative behaviours are driven by the expected information gain. It is important to note that curiosity is necessarily time-dependent, because the expected information gain decreases as the agent resolves uncertainty about its environment; thereby increasing its posterior confidence about hidden or latent states in its generative model. In what follows, we will explore time-dependent fluctuations in curiosity in the context of variations in the precision of prior preferences (i.e., subjective reward or utility)."

Note that you are talking about active inference not free energy principle; however, I don't think you need to worry about this too much.

Please delete the sentence "Thus there was not even the idea that animals irrationally make decisions depending upon the reward and curiosity conflicts." Please replace

this sentence with something like the following:

"In active inference there is no notion of rational versus irrational behaviour. This follows from the complete class theorem [10] that says for any pair of reward functions and choice behaviours, there are some prior beliefs that render the behaviour Bayes optimal. In active inference, these prior beliefs are the prior preferences above. This means that, in principle, one can always explain any choice behaviour in terms of prior preferences that contextualise curiosity. We will show that it is possible to estimate the implicit beliefs, curiosity and confidence of an agent from choice behaviour, under active inference."

In the subsequent paragraph, please assign your "meta-parameter" to the precision of prior preferences and then refer to it as such.

In the description of your generative model and active inference scheme. I would avoid using the word "reward" too much and do not use the symbol r . r does not appear in any of the formal derivations later and can be confusing for people in active inference. For example, it would be simpler to say that "reward will be read as the log prior preference for an outcome that is present, relative to one that is absent. These outcomes depend upon which bandit to choose and their associated latent states (i.e., payoff probabilities)". You later refer to "reward observation". I think this is much easier to understand.

Based on previous experience with this kind of modelling, you may encounter an interesting distinction between the behaviour of agents who do and do not have a generative model of volatility. In other words, you can look at the differences between agents that estimate the changes in w over time, versus those kinds of agents that think w is not changing. This introduces the notion of Bayes optimal forgetting in volatile environments that is often expressed in terms of the balance between exploration and exploitation in active inference.

Please do not call "expected free energy" an "expected utility". This is because it is only the expected reward part of expected free energy that can be read as utility. I am thinking here of expected utility theory in economics. You have gone beyond expected utility because you are including the expected information gain. Please replace expected utility with expected free energy throughout.

Similarly, later in the manuscript, can you remove "which has been difficult to define because of ambiguity". I suggest this because ambiguity has a specific and technical meaning in the present setting. It is possible to rearrange expected free energy into a mixture of risk and ambiguity, where risk is the KL divergence between the posterior predicted outcomes under an action and preferred outcomes. Ambiguity in this setting becomes a conditional entropy given latent states.

At the bottom of page 7, spell out SSM (state space model) when you first mention the "the agent-SSM".

At the bottom of page 8, I would say: "Therefore iFEP is in a position to provide efficient estimators of belief updating to clarify decision-making processing..."

When you introduce the notion of iFEP, it would be scholarly to refer to versions of

this approach in terms of computational phenotyping; for example, [11, 12]. There is also an earlier literature called "observing the observer", which is sometimes referred to as "meta-Bayesian inference".

In the discussion, you might consider replacing the sentence "Consequently their computational performance is restricted..." With:

"In the present context, bounded rationality takes on a different meaning. In active inference, decisions are based upon a generative model under which inferences are optimised using a variational bound on model evidence. This (free energy) bound naturally imposes constraints on the computational resources and time needed to make quick decisions in a Bayes optimal fashion. This is because free energy can be decomposed into accuracy and complexity, where complexity scores both the computational and thermodynamic cost of belief updating. In other words, to be Bayes optimal is to be statistically and metabolically efficient. Furthermore, in active planning as inference, one is trying to minimise the time or path integral of (expected) free energy. Expected free energy is therefore minimised when decisions are made quickly – or when there is a perceived need or prior preference for quick decisions."

Note that the complete class theorem means that everything is Bayes optimal. This means the important questions are about priors and preferences.

With the exceptions of the nomenclature and mathematical formulation above, I thought your description of the generative model, its inversion and use as an observation model were excellent. I also thought that your graphics are informative.

It should be clear from the above that I do not anticipate reviewing this paper unless you move your meta-parameter from curiosity to reward. If you do this, there will be no need to talk about negative curiosity. You should just be able to talk about behaviours that are, and are not, dominated by very precise prior preferences under curiosity.

I hope that these comments help should any revision be required.

1. Schwartenbeck, P. and K. Friston, Computational Phenotyping in Psychiatry: A Worked Example. *eNeuro*, 2016. 3(4).
2. Attias, H. Planning by Probabilistic Inference. in *Proc. of the 9th Int. Workshop on Artificial Intelligence and Statistics*. 2003.
3. Botvinick, M. and M. Toussaint, Planning as inference. *Trends Cogn Sci.*, 2012. 16(10): p. 485-8.
4. Kaplan, R. and K.J. Friston, Planning and navigation as active inference. *Biol Cybern*, 2018. 112(4): p. 323-343.
5. Matsumoto, T. and J. Tani, Goal-Directed Planning for Habituated Agents by Active Inference Using a Variational Recurrent Neural Network. *Entropy*, 2020. 22: p. 564.
6. Lanillos, P., et al. Active Inference in Robotics and Artificial Agents: Survey and Challenges. 2021. arXiv:2112.01871.
7. Lindley, D.V., On a Measure of the Information Provided by an Experiment. *Ann. Math. Statist.*, 1956. 27(4): p. 986-1005.

8. MacKay, D.J.C., Information-Based Objective Functions for Active Data Selection. *Neural Computation*, 1992. 4(4): p. 590-604.
9. Berger, J.O., *Statistical decision theory and Bayesian analysis*. 2011, New York; London: Springer.
10. Wald, A., An Essentially Complete Class of Admissible Decision Functions. *The Annals of Mathematical Statistics*, 1947. 18(4): p. 549-555.
11. Smith, R., et al., Greater decision uncertainty characterizes a transdiagnostic patient sample during approach-avoidance conflict: a computational modelling approach. *Journal of Psychiatry & Neuroscience*, 2021. 46(1): p. E74-E87.
12. Smith, R., et al., Long-term stability of computational parameters during approach-avoidance conflict in a transdiagnostic psychiatric patient sample. *Scientific Reports*, 2021. 11(1).

Reviewer #2 (Remarks to the Author):

**** General Comments ****

The paper proposes a decision-making model for a two-choice task derived from the free energy principle, which integrates an estimation process for probabilistic reward and an active inference process for action selection. By regarding the information gain as the curiosity, the stochastic action selection rule is defined by the Boltzmann distribution in which the utility of each action is given by the weighted sum of the expected reward and the information gain. Using this model, the paper also proposes a sequential Bayesian estimation method to decode the temporal dynamics of the curiosity of the animals from their behavioral data.

The results presented here give interesting insights: 1) The model can address the uncertainty of the external world and the reward acquisition in a dynamic environment; 2) The model follows the free energy principle (FEP) attracting recent attention as a possible unified brain theory; and 3) The dynamic change in the curiosity can be estimated from the time-series of animal's behavioral data. While the presented model is a straightforward integration of two existing ideas: FEP-based curiosity defined in (Schwartenbeck et al., 2019) and non-stationary meta-parameters modeled in (Samejima et al., 2005), the resulting model is quite hopeful for investigating neural substrates associated with curiosity. However, there are several critical concerns and incorrectness in the current manuscript. In my opinion, a substantial revision, based on the specific comments given below, is needed to make this manuscript suitable for publication.

**** Specific Comments ****

1. In the proposed model, the action selection rule (Eq. (3)) comes out of nowhere and is isolated from FEP. Can it be derived from the minimization process of the variational free energy? Otherwise, the utility function defined as the weighted sum of expected reward and the information gain seems heuristic as done in existing studies (e.g., (Houthoofd et al., 2016)). Nonetheless, can we say that the whole procedures in the proposed model follow FEP?

2. I agree that the proposed iFEP can successfully decode the temporal dynamics of the curiosity if the agent perfectly follows the ReCU model (as supported by Fig. 5). However, it is not guaranteed that real animals always make decision according to the ReCU model. This leads to a question whether the quantity and the weight of curiosity estimated by iFEP really reflect internal states of the animals or. In addition, I wonder if it is impossible in two-choice tasks to recognize whether irrational actions were selected by passive random choice or curiosity based on the information gain in FEP (though it may become possible in three-choice tasks by investigating associations two irrational actions with their information gains). Considering those concerns, it is strongly recommended to show that the proposed iFEP model is significantly superior to existing models that can consider the dynamics of the meta-parameters such as (Samejima et al., 2005), in terms of explainability of observed behavioral data. If so, the reliability of FEP/iFEP should be more improved.

3. Fig. 6 shows only the results for a specific rat, raising a concern if iFEP has a higher generalization ability than existing models for behavioral data analysis. At least, it is highly recommended to discuss in what case iFEP should be appropriate or inappropriate.

4. Many hyperparameters to specify the prior distribution for iFEP are missing (e.g., $\epsilon, m_*, v_*, \mu_0, \sigma_\mu$, etc.). In particular, ϵ is very critical for the purpose of this study, because how much the intensity of the curiosity is fluctuated over time. Please specify those values or show the basic policy to tune those values. Also, it is highly recommended to discuss how much the setting of ϵ affects the estimation result.

5. Several equations seem erroneous as shown below. Please correct them.

5-1. Three lines before Eq. (1): "updated belief $P(w_{(i,t)}) = \text{previous belief } P(w_{(i,t-1)}) \times o_t$ " does not make sense. More specifically, if the reward is absent at time t , o_t should be 0 according to the definition in Methods Section, resulting in $P(w_{(i,t)}) = 0$. It is obviously inconsistent with Eqs. (1) and (2). Please revise the statement correctly.

5-2. The 3rd equation from the bottom of page 11: The first term (including the integral) in the right-hand side of $-\ln P(o_t | o_{(1:t-1)}) = \int \dots dw_t - \text{KL}[\dots]$ seems strange. Please correct it.

5-3. Equation (12): The left- and right-hand sides should be the same, but not in the current manuscript. Please correct it.

6. Fig. 2: The time course of (B') seems inconsistent with those of the other panels; thus, I wonder if the colors was wrongly reversed. Please make sure and explain the reason in the response letter if it is correct.

7. Fig. 4: Is the label explaining c_* really "observation"? To me, it seems to be latent variables of unobservable intensity of the curiosity of the agent. Please change it into a correct label or explain the reason in the response letter if it is correct.

** References **

Houthoofd, R., Chen, X., Chen, X., Duan, Y., Schulman, J., de Turck, F., & Abbeel, P.

(2016). VIME: Variational Information Maximizing Exploration. In D. Lee, M. Sugiyama, U. Luxburg, I. Guyon, & R. Garnett (Eds.), *Advances in Neural Information Processing Systems* (Vol. 29). Curran Associates, Inc. <https://proceedings.neurips.cc/paper/2016/file/abd815286ba1007abfbb8415b83ae2cf-Paper.pdf>

Samejima, K., Doya, K., Ueda, Y., Kimura, M., Doya, K., & Kimura, M. (2005). Representation of action-specific reward values in the striatum. *Science*, 310(5752), 1337–1340. <https://doi.org/10.1126/science.1115270>

Schwartenbeck, P., Passacker, J., Hauser, T. U., Fitzgerald, T. H. B., Kronbichler, M., & Friston, K. J. (2019). Computational mechanisms of curiosity and goal-directed exploration. *ELife*, 8, 1–45. <https://doi.org/10.7554/eLife.41703>

Author Rebuttal to Initial comments

Response to Editor's comments

Your manuscript "Decoding reward–curiosity conflict in decision-making from irrational behaviors" has now been seen by 2 referees, whose comments are appended below. You will see that while they find your work of interest, they have raised points that need to be addressed before we can make a decision on publication. The referees' reports seem to be quite clear. Naturally, we will need you to address all of the points raised.

While we ask you to address all of the points raised, the following points need to be substantially worked on:

- Reviewer #1 has mentioned that due to the focus on time sensitive curiosity, you may have accidentally missed the important dynamic ranges in which reward and curiosity trade off against each other. To counter this, please introduce time sensitivity by changing the precision of the reward preferences.

In the revised manuscript, we tried to introduce time-dependent meta-parameter on reward preference, but its estimation from the rat behavioral data led to unnatural interpretation. Please see our response to **Comment 4 by reviewer 1**.

- Please carry out the analyses and iFEP but applying c to expected reward, where the log preferences for the presence or absence of an outcome are one natural unit or zero.

In rebuttal response, we explained that our and suggested representations for the presence and absence of reward are mathematically identical. Please see our response to **Comment 4 by reviewer 1**.

- Please fit the alpha parameter to improve interpretability of results.

In the revised manuscript, we estimated inverse temperature β (call alpha by the reviewer 1). Please see our response to **Comment 10 by reviewer 1**.

- As requested by Reviewer #2, please show that the proposed iFEP model is significantly superior to existing models that can consider the dynamics of the meta-parameters such as (Samejima et al., 2005), in terms of explainability of observed behavioral data.

In the revised manuscript, we compared iFEP and Q-learning, and showed that our iFEP approach on the ReCU model is superior to Q-learning for full explanation of the rat behavior. Please see our response to **Comment 2 by reviewer 2**.

- Please derive Equation 3 from the minimization process of the variational free energy.

In the revised manuscript, we derived equation 3. Please see our response to **Comment 1 by reviewer 2**.

- Please discuss if iFEP has a higher generalization ability than existing models for behavioral data analysis.

In the revised manuscript, we discussed this point. Please see our response to **Comment 3 by reviewer 2**.

- Please specify the hyperparameters values or show the basic policy to tune those values. In addition to these points, it would also be beneficial to address the following concerns:

In the revised manuscript, we clearly mentioned what is the hyperparameters and how to estimate these. Please see our response to **Comment 4 by reviewer 2**.

Response to Reviewer 1's comments

Comment 1

I enjoyed reading this sophisticated numerical and empirical study of choice behaviour under the free energy principle. In terms of the ambition and potential importance of this work, I thought this was a substantial contribution. If you can develop an inverse FEP procedure to estimate belief updating, confidence and curiosity in animal studies, then this opens the door to potentially important applications in translational (e.g., pharmacological) studies. I was also impressed by your mathematical fluency in translating the free energy principle into your particular generative model – and the application of variational procedures (and particle filtering) to estimate hidden belief states of experimental subjects.

I would like to thank the reviewer for carefully reading our manuscript and providing valuable feedback.

Comment 2

Although I think it is important that you bring this work to closure, there are some foundational issues you need to address first. I suspect that these will require a simplification of your model and a reanalysis of the empirical data. This is because your extension to the FEP is not licensed or necessary.

We apologize for not being able to fully understand the reviewer's intention. We guessed that "simplification of your model" means that introducing the meta-parameter on the reward (i.e., prior preference), instead of information gain, as also suggested in **Comment 4**. In the revised manuscript, we performed the suggested analysis on the rat behavioral data. Please see our response to **Comment 4**.

Comment 3

Because you have focused on time sensitive curiosity, as opposed to preferences, you have accidentally missed the important dynamic ranges in which reward and curiosity trade off against each other. Specifically, what you have studied is called "active inference" that can be derived from the free energy principle. The free energy principle is a first principle account based upon statistical physics. For your interest, you can read about where the expected free energy comes from in this recent summary: <https://arxiv.org/ftp/arxiv/papers/2201/2201.06387.pdf>

> *Because you have focused on time sensitive curiosity, as opposed to preferences, you have accidentally missed the important dynamic ranges in which reward and curiosity trade off against each other.*

I agreed that if the agent is Bayes optimal, i.e., the meta-parameter of curiosity $c=1$, reward and curiosity should be a trade-off. As the reviewer expected, we showed it at Bayes optimal condition in the revised manuscript (**Extended Data Fig. 2**, see **Lines 193-197**).

> Specifically, what you have studied is called “active inference” that can be derived from the free energy principle. The free energy principle is a first principle account based upon statistical physics. For your interest, you can read about where the expected free energy comes from in this recent summary: <https://arxiv.org/ftp/arxiv/papers/2201/2201.06387.pdf>

In the revised manuscript, we clearly mentioned that we modeled “active inference” based on the free energy principle (see **Lines 65-67**) and that the free energy principle is a first principle account based upon statistical physics by Prof. Friston (see **Lines 63-65**).

Comment 4

The key point here is that the functional form of expected free energy (something you have called expected utility) cannot be changed. However, you can introduce a time sensitivity by changing the precision of the reward preferences. In other words, instead of applying c to the expected information gain, you can apply it to the reward preferences. This means that c now plays the role of a precision of preferences about outcomes.

If you take this approach, you will be able to fit your empirical data more sensibly.

> The key point here is that the functional form of expected free energy (something you have called expected utility) cannot be changed.

The reviewer expressed concern that the functional form of expected utility (negative expected free energy) cannot be changed. However, we think there is no such rule. Actually, Houthoof et al., 2016, mentioned Comment 1 by another reviewer, modeled reinforcement learning of information maximizing exploration by introducing the curiosity meta-parameter to the information gain, as done by us. Because Houthoof et al., 2016 have been cited in more than 650 papers, this idea must be common. In the revised manuscript, we clearly mentioned Houthoof et al., 2016 (see **Lines 518-519**).

In addition, we agree that the expected utility we called ('expected net utility' we call in the revised manuscript in response to Suggestion 6 below) is negative expected free energy developed by Prof. Friston. Because we preferred to formulate decision-making by maximizing weighted sum of reward and information gain, we used the expected utility. In the revised manuscript, we mentioned this point (see **Lines 172-174**).

> However, you can introduce a time sensitivity by changing the precision of the reward preferences. In other words, instead of applying c to the expected information gain, you can apply it to the reward preferences.

This means that c now plays the role of a precision of preferences about outcomes.

As suggested, we introduced the meta-parameter to the reward preference and applied it to the rat behavioral data. However, the estimated time-series of the reward meta-parameter cannot be clearly interpreted. In the revised manuscript, we created a new section 'Evaluations of alternative models from rat behaviors' to include this analysis (**Extended Data Fig. 7**; see **Lines 346-356**).

Comment 5

The problem with the current simulations — and the empirical estimates — is that c can be negative. However, the expected information gain is nonnegative. This means that you are explaining behaviour in terms of an aversion to expected information gain. This is a consequence of specifying reward preferences as a logit function (your first equation). The problem with the logit function is that the important variations in preference are limited to an extremely small range; namely between $P_0 = 0.99 - 1$.

> The problem with the current simulations — and the empirical estimates — is that c can be negative. However, the expected information gain is nonnegative. This means that you are explaining behaviour in terms of an aversion to expected information gain.

The reviewer is concerned with our assumption that the curiosity meta-parameter c can be a negative, which can explain aversive behavior to information gain. In the revised manuscript, we proposed a new analysis to distinguish the positive and negative curiosity (**Fig. 2H-J and Lines 217-231**). With this analysis on the actual behavioral data, we further validated that the rat had the negative curiosity (**Fig. 6A-C and Lines 322333**).

In addition, we believe that the negative value of the curiosity meta-parameter c is biologically reasonable, because autism spectrum disorder (ASD) patients have been known to avoid new information. In revision, we mentioned this point (see **Lines 50-53 and Lines 244-246**).

> This is a consequence of specifying reward preferences as a logit function (your first equation).

The reviewer considered that reward given by a logit function is related to the negative value of the curiosity meta-parameter. However, mathematically speaking, two representations for presence and absence of reward: ' $\ln\{P_o/(1-P_o)\}$ and 0' and ' $\ln P_o$ and $\ln (1-P_o)$ ' are identical, because the agent selection is based on the relative value of rewards, not absolute values. Here, the left-right difference of the expected reward was calculated below, which confirmed that usages of log preference and logit function both lead to the identical decision-making.

In usage of logit function: ' $\ln\{P_o/(1-P_o)\}$ vs. 0'

$$\Delta E[R] = [f(\mu_l) \ln\{P_o/(1-P_o)\} + \{1-f(\mu_l)\} 0] - [f(\mu_r) \ln\{P_o/(1-P_o)\} + \{1-f(\mu_r)\} 0] = \{f(\mu_l) - f(\mu_r)\} \ln\{P_o/(1-P_o)\}$$

In usage of log preference: ' $\ln P_o$ vs. $\ln (1-P_o)$ '

$$\Delta E[R] = [f(\mu_l) \ln P_o + \{1-f(\mu_l)\} \ln (1-P_o)] - [f(\mu_r) \ln P_o + \{1-f(\mu_r)\} \ln (1-P_o)] = \{f(\mu_l) - f(\mu_r)\} \ln\{P_o/(1-P_o)\}$$

where $f(\mu_l)$ and $f(\mu_r)$ indicate the predicted reward probabilities of the left and right options, respectively. In both equations, the first and second terms represents the expected reward from the left and right options, respectively.

> The problem with the logit function is that the important variations in preference are limited to an extremely small range; namely between $P_o = 0.99 - 1$.

The reviewer is concerned about the dynamic range of the logit function. As shown in **Extended Data Fig. 1**, the value almost linearly increases with P_o from 0 to 0.9. Thus, we think that the P_o less than 0.9 can exhibit important variations of the reward. As indicated by the reviewer, in the region of $P_o = 0.99-1$, the value exponentially increases. In the revised manuscript, we included this plot (see **Extended Data Fig. 1**, see **Lines 119-122**).

Comment 6

The key insight here is that expected information gain and expected reward in active inference have natural units (when using natural logarithms). This means a strong preference requires a difference in log probabilities of about 2 to 3. The logit function only produces these differences with values of P_0 that are close to one. Furthermore, the logic function places an upper bound on P_0 that is unnecessary with no precedent for this functional form.

As mentioned in our response to Comment 5, the usage of the logit function representing intensity of reward is essentially the same as the usage of log preference reward ($\ln P_0$ and $\ln (1-P_0)$). Thus, we still prefer to give the presence and absence of reward as 0 and $\ln P_0 - \ln (1-P_0)$, which is the logit function, because of three reasons. First, the relative value of rewards, not absolute value, affects decision-making. Second, $\ln P_0$ and $\ln (1-P_0)$ take negative values, although most readers regard rewards to be positive values. Third, the agent obtains 'reward value' of $\ln (1-P_0)$ even with no observation of actual reward, which is odd. In the revised manuscript, we mentioned that the usage of the logic function is comparable to the log preference reward (see Lines 119-122).

Comment 7

Could you redo your analyses and iFEP but applying c to expected reward, where the log preferences for the presence or absence of an outcome are one natural unit or zero. Expected free energy = Expected information gain + Expected reward

$$\text{Expected reward} = E[\log p(o_{t+1})] = c \cdot E[\log p_0(o_{t+1})]$$

$$\begin{aligned} \text{Prior preferences} &= p(o_{t+1} = \text{absent}) = 1 - P_0^c \\ &= p(o_{t+1} = \text{present}) = P_0^c \end{aligned}$$

Where $P_0 = 1$ is a baseline preference and c is a meta-parameter that sets the precision prior preferences.

> Could you redo your analyses and iFEP but applying c to expected reward, where the log preferences for the presence or absence of an outcome are one natural unit or zero.

This response to this comment is the same as Comment 4.

> Where $P_0 = 1$ is a baseline preference and c is a meta-parameter that sets the precision prior preferences.

When $P_0=1$, $\ln(1-P_0)=-\infty$. Thus, we assumed P_0 less than 1.

Comment 8

You should then find an interesting switch between curiosity and reward with values of c that are around the expected information gain. I notice that your expected information gain was very small. This may be because you are using a continuous hidden state and a Gaussian or Laplace approximation to the posterior. It would be useful to start off with a prior over the hidden state w so that the entropy is nontrivial (e.g., 5) natural units greater than the posterior entropy after a typical number of trials (e.g., 32).

This is an interesting suggestion to see switching between the expected reward and the expected information gain. As indicated by the reviewer, in the previous simulation, the expected information gain was much smaller than the expected reward. In the revised manuscript, we changed the parameters and observed switching behavior in simulation, as the reviewer suggested, to change mean and variance of recognition prior (see **Extended Data Fig. 2A and C**, see **Lines 193-196, Lines 209-211**).

Comment 9

What you should see is that as the simulated rat becomes more confident about the values of w , the expected information gain falls by a nontrivial amount. At present, it looks as if expected information gain varies by a trivial amount over several hundred trials, which means something is wrong with your setup. In other words, it looks as if the (synthetic) rats are not really learning anything about the payoff probabilities. In consequence, the differences in expected free energy between different actions may be very small and therefore it will be difficult to estimate the belief updating from observed choice behaviour.

This comment is related to **Comment 8**.

> What you should see is that as the simulated rat becomes more confident about the values of w , the expected information gain falls by a nontrivial amount.

Regarding Figure 2, the reviewer is concerned that the agent did not learn the reward probabilities, because the expected information gain did not significantly fall during learning. As responded to

comment 8, the expected information gain significantly falls at early phase (see Extended Data Fig. 2A and C, see Lines 193-196, Lines 209-211).

> At present, it looks as if expected information gain varies by a trivial amount over several hundred trials, which means something is wrong with your setup.

The reviewer is also concerned that the expected information fluctuated trial-by-trial and did not converge (Fig. 2E and E'). However, this is normal, not strange. In our model, the agent's posterior distribution of the latent variable w of one option blurred with not sampling of the option, because the agent assumed random walk of w . Thus, the expected information gain consequently varied in time due to biased probabilistic selection of options. In the revised manuscript, we mentioned this point (see Lines 187-191).

> In other words, it looks as if the (synthetic) rats are not really learning anything about the payoff probabilities.

We showed that the recognized reward probabilities converged to their ground truths. This is the evidence that the agent correctly learns to recognize the reward probabilities (Fig. 2C and C').

Comment 10

A crucial point here, is that you have to decide whether action is sampled from the posterior beliefs about choices or whether you select the most likely action. Usually, when using active inference to fit choice behaviour in humans, one uses a further softmax precision or inverse temperature parameter (called alpha) that is applied to the expected free energy before applying the softmax operator. This allows you to fit a subject-specific stochasticity to action selection and obtain veridical estimates of the parameters that underwrite belief updating.

According to this suggestion, we introduced inverse temperature into the softmax action selection (see equation 3, Lines 178-180). In the revised manuscript, we also estimated the inverse temperature (see Lines 674-683).

Note that inverse temperature is denoted by β , not α , in the revised manuscript, because α has already been used for learning rate in equation (1) and β is widely used for the inverse temperature in statistical physics.

Comment 11

You appear not to have fitted this (α) parameter and therefore I think it is difficult to interpret your results; especially when allowing for a random walk on c . I do not think you need to model random walks or volatility on either the expected information gain or the reward preferences. This is because active inference explains time-dependent changes in the balance between curiosity and reward. This follows because as exploration increases confidence about latent states (i.e., w) the expected information gain falls to 0 and curiosity gives way to exploitative, preference-satisfying behaviour.

> You appear not to have fitted this (α) parameter

In the revised manuscript, we estimated the inverse temperature parameter, as mentioned in our response to Comment 10.

> I think it is difficult to interpret your results; especially when allowing for a random walk on c . I do not think you need to model random walks or volatility on either the expected information gain or the reward preferences.

We think that the agents adaptively tune the curiosity meta-parameter in a context-dependent manner. In fact, we applied our inverse approach (iFEP) to the rat behaviors and demonstrated that the estimated level of curiosity increased at which the reward probabilities suddenly changed. This result can be clearly interpreted such that the rat recognized the rule change and adaptively controlled the extent to which the agent sought new information (Fig. 5H).

To support this statement in the revised manuscript, we further analyzed the estimated time-series of the curiosity meta-parameter and expected information gains. We found that temporal change in the curiosity meta-parameter correlated with the sum of the expected information gains between two options (Fig. 6F, G), suggesting that the rat actively controlled the curiosity depending on current recognition and its confidence. In the revised manuscript, we created a new section to mention this point (see Lines 334343).

Comment 12

This is a fundamental aspect of active inference that you would expect to see in your simulations and in the empirical data that you analyse. If the empirical data is driven largely by preferences with minimal exploration, then this should be explained sufficiently by having a large value of c applied to the baseline prior preferences above. In other words, adding time-dependent precisions to expected free energy is interesting but may make it difficult for you to invert the active inference (FEP) model without introducing lots of conditional independencies. I know that this is the case because there is quite a lot of experience of doing this in the field of active inference; particularly, in computational phenotyping in psychiatry. An early example can be found here [1].

In this comment, the reviewer re-stated the content of **Comment 11**, indicating that the introduction of time-dependent meta-parameter is not suited for understanding the active inference. Rather, we believe that our approach is able to deepen our understanding of the active inference.

The active inference proposed by Prof. Friston assumed a rational, optimal agent, who wants to know the uncertain environment for finally exploiting the rewards. However, we think that all living organisms including us are not always rational; they are sometimes motivated to just know the situation, irrespective of the rewards. In fact, a previous machine learning study (Houthoofd et al., 2016) formulated such an urge to explore new information by introducing the curiosity meta-parameter to the expected information gain. This idea was widely accepted, and in fact, the study of Houthoofd et al. has been cited by 645 papers.

In other aspects, the real environment is not stationary, but dynamically changing. Thus, the agents face an inevitable dilemma and conflict between reward and curiosity. Therefore, decoding time-dependent such conflict from behavioral data is important for understanding active inference of living organisms in a realistic dynamic environment. Actually, our iFEP approach actually decoded the time-dependent curiosity meta-parameter and found that the curiosity increased at the timing of sudden changes in the reward probabilities in the two-choice task (**Fig. 5H**). Furthermore, in the revised manuscript, we newly that the temporal derivative of the curiosity is positively correlated with the information gain (e.g., uncertainty of recognition) (**Fig. 6F, G**), suggesting that the rat autonomously controlled the level of curiosity in a contextdependent manner.

Suggestions for the paper:

Suggestion 1

Please replace the sentences in the introduction starting with "Furthermore, FEP was.... between reward and curiosity." With something like:

“The free energy principle is a first principle account of sentient behaviour that is motivated from the statistical physics of self organisation. From the perspective of behavioural science and decision-making, the most interesting contribution of the FEP is an expected free energy that scores the likelihood of an action in the setting of active (planning as) inference [2-6]. The ensuing formulation of action and perception, in terms of active inference, accommodates reward learning by formulating rewards as preferences over outcomes. This allows one to express both the information-seeking and preference-seeking contributions to expected free energy in the same units (usually natural units when using natural logarithms). This rests upon the decomposition of expected free energy into expected information gain and expected reward. These two components underwrite optional Bayesian design [7, 8] and decision theoretic [9] formulations of behaviour respectively.

Crucially, the balance between the two components depends upon the precision of the prior preferences that underwrite subjective reward or utility. Agents with very precise prior preferences will engage in more exploitative behaviour; whereas — in the absence of prior preferences — explorative behaviours are driven by the expected information gain. It is important to note that curiosity is necessarily time-dependent, because the expected information gain decreases as the agent resolves uncertainty about its environment; thereby increasing its posterior confidence about hidden or latent states in its generative model. In what follows, we will explore time-dependent fluctuations in curiosity in the context of variations in the precision of prior preferences (i.e., subjective reward or utility)."

Note that you are talking about active inference not free energy principle; however, I don't think you need to worry about this too much.

> From the perspective of behavioural science and decision-making, the most interesting contribution of the FEP is an expected free energy that scores the likelihood of an action in the setting of active (planning as) inference [2-6].

In the revised manuscript, we explain the expected free energy with citing the suggested papers [2-4] in Introduction (see **Lines 68-69**). The suggested papers [5, 6] were cited in Discussion, as these two papers were appropriate to mention as the application of expected free energy to robotics in Discussion (see **Lines 387-388**).

> These two components underwrite optional Bayesian design [7, 8] and decision theoretic [9] formulations of behaviour respectively.

In the revised manuscript, we cited the suggested papers [7-9] in Introduction (see Line 67)

> Crucially, the balance between...

In this paragraph which the reviewer suggested us to replace with part of Introduction, the reviewer again recommended us to introduce the time-dependent meta-parameter on the expected reward (prior preference the reviewer says), as mentioned in comment 4. Accordingly, we revised the manuscript as suggested. Please see our response to Comment 4.

> Agents with very precise prior preferences will engage in more exploitative behaviour; whereas — in the absence of prior preferences — explorative behaviours are driven by the expected information gain.

In the revised manuscript, we mentioned this point in Methods (see Lines 544-551).

Suggestion 2

Please delete the sentence "Thus there was not even the idea that animals irrationally make decisions depending upon the reward and curiosity conflicts."

Please replace this sentence with something like the following:

"In active inference there is no notion of rational versus irrational behaviour. This follows from the complete class theorem [10] that says for any pair of reward functions and choice behaviours, there are some prior beliefs that render the behaviour Bayes optimal. In active inference, these prior beliefs are the prior preferences above. This means that, in principle, one can always explain any choice behaviour in terms of prior preferences that contextualise curiosity. We will show that it is possible to estimate the implicit beliefs, curiosity and confidence of an agent from choice behaviour, under active inference."

It seems that the reviewer strongly suggested us not to address irrational behavior. However, our scope was the irrationality of decision-making depending on the reward and curiosity conflict, and this is an important assumption and core of our inverse approaches. In our response to Comment 12, we explained this in detail.

Suggestion 3

In the subsequent paragraph, please assign your "meta-parameter" to the precision of prior preferences and then refer to it as such.

This comment is the same as comment 4 and suggestion 1. According to this suggestion, we introduced the time-dependent meta-parameter on the expected reward as a subjective reward (prior preference the reviewer says). We found that the subjective reward estimated from the rat behavioral data dynamically changed and was sometimes close to zero. However, this result must be odd because the rat in this experiment was very hungry and should crave for food. Please see our responses to **Comment 4** for details.

Suggestion 4

In the description of your generative model and active inference scheme. I would avoid using the word "reward" too much and do not use the symbol r . r does not appear in any of the formal derivations later and can be confusing for people in active inference. For example, it would be simpler to say that "reward will be read as the log prior preference for an outcome that is present, relative to one that is absent. These outcomes depend upon which bandit to choose and their associated latent states (i.e., payoff probabilities)". You later refer to "reward observation". I think this is much easier to understand.

The reviewer avoids using the word "reward", but prefers the word "prior preference". That is exactly Prof. Friston's way. However, we still prefer to use the word "reward", as used in reinforcement learning which was formulated for maximizing the reward. We think that choice of words (i.e., reward vs. prior preference) might depend on personal orientation, because both are mathematically the same. Actually, Houthoofd et al., 2016 (citations > 650) used the terminology "reward" along with the expected information gain, whose formulation is identical to our manuscript.

We also explained why we used a logit function for the reward in our response to **Comment 11**.

Suggestion 5

Based on previous experience with this kind of modelling, you may encounter an interesting distinction between the behaviour of agents who do and do not have a generative model of volatility. In other words, you can look at the differences between agents that estimate the changes in w over time, versus those kinds of agents that think w is not changing. This introduces the notion of Bayes optimal forgetting in volatile environments that is often expressed in terms of the balance between exploration and exploitation in active inference.

Thank you for the informative comment. The reviewer suggested a comparison between our ReCU model with a volatile (dynamic) environment and a model assuming a constant environment. In the revised manuscript, we mentioned that equations (1, 2) of recognition update becomes the case assuming a constant environment when $p_w = \text{inf}$, $\sigma = 0$ (see Lines 144-145). In iFEP, we also estimated σ from the rat behavioral data as 0.56, suggesting that the rat assumed a dynamic environment. In the revised manuscript, we mentioned this point (see Lines 314-321).

Suggestion 6

Please do not call “expected free energy” an “expected utility”. This is because it is only the expected reward part of expected free energy that can be read as utility. I am thinking here of expected utility theory in economics. You have gone beyond expected utility because you are including the expected information gain.

Please replace expected utility with expected free energy throughout.

We did not know that there is ‘expected utility theory’ in economics. Thank you for pointing this out. In revision, we renamed it as ‘expected net utility’ to avoid confusion (see Line 167).

We would prefer not to call it an ‘expected free energy,’ if at all possible. This is because for most people it could be more natural that the agent tries to maximize total reward, and less natural that the agent minimizes the expected free energy. This could be our personal preference, because the expected free energy is essentially identical to the expected net utility except its sign. In the revised manuscript, we clearly mentioned that the ‘expected net utility’ is negative of the ‘expected free energy’ (see Lines 172-174).

Suggestion 7

Similarly, later in the manuscript, can you remove "which has been difficult to define because of ambiguity". I suggest this because ambiguity has a specific and technical meaning in the present setting. It is possible to rearrange expected free energy into a mixture of risk and ambiguity, where risk is the KL divergence between the posterior predicted outcomes under an action and preferred outcomes. Ambiguity in this setting becomes a conditional entropy given latent states.

Following this comment, we removed the indicated sentence.

Suggestion 8

At the bottom of page 7, spell out SSM (state space model) when you first mention the "the agent-SSM".

As suggested, we used 'SSM' instead of 'the agent-SSM'.

Suggestion 9

At the bottom of page 8, I would say: "Therefore iFEP is in a position to provide efficient estimators of belief updating to clarify decision-making processing..."

In the revised manuscript, we modified this sentence as suggested (see Lines 299-300).

Suggestion 10

When you introduce the notion of iFEP, it would be scholarly to refer to versions of this approach in terms of computational phenotyping; for example, [11, 12]. There is also an earlier literature called "observing the observer", which is sometimes referred to as "meta-Bayesian inference".

Thank you for informing us about previous computational phenotyping approaches, which should be related to our iFER. In the revised manuscript, we mentioned and cited these suggested papers (see Lines 263-265).

Suggestion 11

In the discussion, you might consider replacing the sentence "Consequently their computational performance is restricted..." With:

"In the present context, bounded rationality takes on a different meaning. In active inference, decisions are based upon a generative model under which inferences are optimised using a variational bound on model evidence. This (free energy) bound naturally imposes constraints on the computational resources and time needed to make quick decisions in a Bayes optimal fashion. This is because free energy can be decomposed into accuracy and complexity, where complexity scores both the computational and thermodynamic cost of belief updating. In other words, to be Bayes optimal is to be statistically and metabolically efficient. Furthermore, in active planning as inference, one is trying to minimise the time or path integral of (expected) free energy. Expected free energy is therefore minimised when decisions are made quickly – or when there is a perceived need or prior preference for quick decisions."

Thank you for pointing this out. The indicated sentence could not be related to the bounded rationality. Therefore, in the revised manuscript, we deleted this sentence.

Suggestion 12

Note that the complete class theorem means that everything is Bayes optimal. This means the important questions are about priors and preferences.

It also appears that the reviewers strongly suggested to treat the agent's behavior as Bayesian optimal. However, we addressed irrational decision-making away from the Bayesian optimum. In our response to **Comment 12**, we explained this in detail.

Suggestion 13

With the exceptions of the nomenclature and mathematical formulation above, I thought your description of the generative model, its inversion and use as an observation model were excellent. I also thought that your graphics are informative.

Thank you for your evaluation.

Suggestion 14

It should be clear from the above that I do not anticipate reviewing this paper unless you move your metaparameter from curiosity to reward. If you do this, there will be no need to talk about negative curiosity. You should just be able to talk about behaviours that are, and are not, dominated by very precise prior preferences under curiosity.

According to this comment, in the revised manuscript, we tried to put the time-dependent meta-parameter on the subjective reward. We found that the subjective reward estimated from the rat behavioral data dynamically changed and sometimes close to zero. However, this result must be odd because the rat in this experiment was very hungry and should want to have food. Please see our response to **Comment 4** in detail.

1. Schwartenbeck, P. and K. Friston, Computational Phenotyping in Psychiatry: A Worked Example. eNeuro, 2016. 3(4).
2. Attias, H. Planning by Probabilistic Inference. in Proc. of the 9th Int. Workshop on Artificial Intelligence and Statistics. 2003.
3. Botvinick, M. and M. Toussaint, Planning as inference. Trends Cogn Sci., 2012. 16(10): p. 485-8.
4. Kaplan, R. and K.J. Friston, Planning and navigation as active inference. Biol Cybern, 2018. 112(4): p. 323-343.
5. Matsumoto, T. and J. Tani, Goal-Directed Planning for Habituated Agents by Active Inference Using a Variational Recurrent Neural Network. Entropy, 2020. 22: p. 564. 6. Lanillos, P., et al. Active Inference in Robotics and Artificial Agents: Survey and Challenges. 2021. arXiv:2112.01871.
7. Lindley, D.V., On a Measure of the Information Provided by an Experiment. Ann. Math. Statist., 1956. 27(4): p. 986-1005.
8. MacKay, D.J.C., Information-Based Objective Functions for Active Data Selection. Neural Computation, 1992. 4(4): p. 590-604.
9. Berger, J.O., Statistical decision theory and Bayesian analysis. 2011, New York; London: Springer.

10. Wald, A., An Essentially Complete Class of Admissible Decision Functions. *The Annals of Mathematical Statistics*, 1947. 18(4): p. 549-555.
11. Smith, R., et al., Greater decision uncertainty characterizes a transdiagnostic patient sample during approach-avoidance conflict: a computational modelling approach. *Journal of Psychiatry & Neuroscience*, 2021. 46(1): p. E74-E87.
12. Smith, R., et al., Long-term stability of computational parameters during approach-avoidance conflict in a transdiagnostic psychiatric patient sample. *Scientific Reports*, 2021. 11(1).

Response to Reviewer 2's comments

Remarks

The paper proposes a decision-making model for a two-choice task derived from the free energy principle, which integrates an estimation process for probabilistic reward and an active inference process for action selection. By regarding the information gain as the curiosity, the stochastic action selection rule is defined by the Boltzmann distribution in which the utility of each action is given by the weighted sum of the expected reward and the information gain. Using this model, the paper also proposes a sequential Bayesian estimation method to decode the temporal dynamics of the curiosity of the animals from their behavioral data.

The results presented here give interesting insights: 1) The model can address the uncertainty of the external world and the reward acquisition in a dynamic environment; 2) The model follows the free energy principle (FEP) attracting recent attention as a possible unified brain theory; and 3) The dynamic change in the curiosity can be estimated from the time-series of animal's behavioral data. While the presented model is a straightforward integration of two existing ideas: FEP-based curiosity defined in (Schwartenbeck et al., 2019) and non-stationary meta-parameters modeled in (Samejima et al., 2005), the resulting model is quite hopeful for investigating neural substrates associated with curiosity. However, there are several critical concerns and incorrectness in the current manuscript. In

my opinion, a substantial revision, based on the specific comments given below, is needed to make this manuscript suitable for publication.

Thank you for reviewing our manuscript and providing valuable suggestions. We prepared responses to specific comments below and revised our manuscript according to the comments.

Comment 1

In the proposed model, the action selection rule (Eq. (3)) comes out of nowhere and is isolated from FEP. Can it be derived from the minimization process of the variational free energy? Otherwise, the utility function defined as the weighted sum of expected reward and the information gain seems heuristic as done in existing studies (e.g., (Houthoof et al., 2016)). Nonetheless, can we say that the whole procedures in the proposed model follow FEP?

> In the proposed model, the action selection rule (Eq. (3)) comes out of nowhere and is isolated from FEP.

Can it be derived from the minimization process of the variational free energy?

Yes. It can be analytically derived, even with heuristic weighting in the expected utility. In the revised manuscript, we showed the derivation of equation (3) (see Lines 512-527).

> Otherwise, the utility function defined as the weighted sum of expected reward and the information gain seems heuristic as done in existing studies (e.g., (Houthoof et al., 2016))

As mentioned by the reviewer, the weighted sum of expected reward and the information gain is heuristic, which is the same as mentioned by Houthoof et al., 2016. Notably, there is no relationship between derivation of action selection rule and heuristic formulation of the expected utility. In the revised manuscript, we mentioned that we adopted heuristic weighting in the expected utility (see Lines 518-519)

Comment 2

I agree that the proposed iFEP can successfully decode the temporal dynamics of the curiosity if the agent perfectly follows the ReCU model (as supported by Fig. 5). However, it is not guaranteed that real animals always make decision according to the ReCU model. This leads to a question whether the

quantity and the weight of curiosity estimated by iFEP really reflect internal states of the animals or. In addition, I wonder if it is impossible in two-choice tasks to recognize whether irrational actions were selected by passive random choice or curiosity based on the information gain in FEP (though it may become possible in three-choice tasks by investigating associations two irrational actions with their information gains). Considering those concerns, it is strongly recommended to show that the proposed iFEP model is significantly superior to existing models that can consider the dynamics of the meta-parameters such as (Samejima et al., 2005), in terms of explainability of observed behavioral data. If so, the reliability of FEP/iFEP should be more improved.

> it is not guaranteed that real animals always make decision according to the ReCU model. This leads to a question whether the quantity and the weight of curiosity estimated by iFEP really reflect internal states of the animals or. I wonder if it is impossible in two-choice tasks to recognize whether irrational actions were selected by passive random choice or curiosity based on the information gain in FEP.

Thank you for raising such a great question. In the revised manuscript, we presented a diagram distinguishing between passive random choice and curiosity-dependent choice (see Lines 217-231). Based on this diagram, we showed that the actual rat did not passively select options but had negative curiosity on average and adaptively tunes its level in response to sudden changes in the reward probabilities (see Lines 334-344).

> Considering those concerns, it is strongly recommended to show that the proposed iFEP model is significantly superior to existing models that can consider the dynamics of the meta-parameters such as (Samejima et al., 2005), in terms of explainability of observed behavioral data.

This is also a great suggestion. In the revised manuscript, from the rat behavioral data, we showed that the temporal derivative of the curiosity was driven by the information gain (e.g., uncertainty of recognition) (Fig. 6D), which gives explainability of a context-dependent modulation of the rat's curiosity. In addition, following Samejima et al., 2005, we adopted Q-learning and estimated time-dependent inverse temperature β controlling passive random choice (Extended Data Fig. 8A-D). We found that the inverse temperature is positively correlated with the information gain (Extended Data Fig. 8 G-H). This result clearly indicates that even if we adopt Q-learning to explain the rat behaviors, we still need 'information gain' in the ReCU model. In revision, we carefully explained this point (Lines 357-371).

In addition, the agent with large positive and negative curiosity actively selects and avoids the certain option, respectively, which can explain pathological decision-making observed from ADHD and ASD patients. These cannot be explained by usual Q-learning. In the revised manuscript, we mentioned the relationship between ADHD/ASD patients and positive/negative curiosity (Lines 244-246, Lines 252-253).

Comment 3

Fig. 6 (Fig. 5 in the revised manuscript) shows only the results for a specific rat, raising a concern if iFEP has a higher generalization ability than existing models for behavioral data analysis. At least, it is highly recommended to discuss in what case iFEP should be appropriate or inappropriate.

According to this comment, in the revised manuscript, we discussed how iFEP is more useful than existing Q-learning model (see Lines 418-429).

Comment 4

Many hyperparameters to specify the **prior distribution** for iFEP are missing (e.g., ϵ , m_* , v_* , μ_0 , σ_μ , etc.). In particular, ϵ is very critical for the purpose of this study, because how much the intensity of the curiosity is fluctuated over time. Please specify those values or show the **basic policy to tune** those values.

Also, it is highly recommended to discuss how much the setting of ϵ affects the estimation result.

We apologize for the missing information. We estimated the parameters using the particle filter based on the self-organizing state-space model. In the revised article, we improved the explanation of the parameter estimation method (see Lines 674-689), listed the values of the hyperparameters and mentioned policy to set those values (see Lines 690-693). Note that we changed the notation of several hyperparameters in the revised manuscript.

Following this suggestion, we also examined the effect of ϵ on the iFEP, and demonstrated that the estimation performance of the curiosity meta-parameter was robust against even 5 times change in ϵ . In the revised manuscript, we explained this point (Extended Data Fig. 4; see Lines 298-299).

Several equations seem erroneous as shown below. Please correct them.

Minor comment 1

Three lines before Eq. (1): “updated belief $P(w_{(i,t)}) = \text{previous belief } P(w_{(i,t-1)}) \times o_t$ ” does not make sense.

More specifically, if the reward is absent at time t , o_t should be 0 according to the definition in Methods Section, resulting in $P(w_{(i,t)}) = 0$. It is obviously inconsistent with Eqs. (1) and (2). Please revise the statement correctly.

We apologize for our poor explanation of the Bayes update rule. In the revised manuscript, we carefully revised the statement (see Lines 151-152).

Minor comment 2

The 3rd equation from the bottom of page 11: The first term (including the integral) in the right-hand side of $-\ln P(o_t | o_{(1:t-1)}) = \int \dots dw_t - KL[\dots]$ seems strange. Please correct it.

Thank you for pointing it out. We have corrected this (see Line 471).

Minor comment 3

Equation (12): The left- and right-hand sides should be the same, but not in the current manuscript. Please correct it.

Thank you for pointing it out. We have corrected this (see Line 645).

Minor comment 4

Fig. 2: The time course of (B') seems inconsistent with those of the other panels; thus, I wonder if the colors was wrongly reversed. Please make sure and explain the reason in the response letter if it is correct.

Thank you for pointing it out. We have corrected this (see Fig. 2).

Minor comment 5

Fig. 4: Is the label explaining c_* really “observation”? To me, it seems to be latent variables of unobservable intensity of the curiosity of the agent. Please change it into a correct label or explain the reason in the response letter if it is correct.

Thank you for pointing it out. We have corrected this (see Fig. 4).

** References **

Houthoofd, R., Chen, X., Chen, X., Duan, Y., Schulman, J., de Turck, F., & Abbeel, P. (2016). VIME: Variational Information Maximizing Exploration. In D. Lee, M. Sugiyama, U. Luxburg, I. Guyon, & R. Garnett (Eds.), *Advances in Neural Information Processing Systems* (Vol. 29). Curran Associates, Inc.

Samejima, K., Doya, K., Ueda, Y., Kimura, M., Doya, K., & Kimura, M. (2005). Representation of action-specific reward values in the striatum. *Science*, 310(5752), 1337–1340.

Schwartenbeck, P., Passecker, J., Hauser, T. U., Fitzgerald, T. H. B., Kronbichler, M., & Friston, K. J. (2019). Computational mechanisms of curiosity and goal-directed exploration. *ELife*, 8, 1–45.

Decision Letter, first revision:

Date: 20th January 23 12:09:54

Last Sent: 20th January 23 12:09:54

Triggered By: Ananya Rastogi

From: ananya.rastogi@nature.com

To: nhonda@hiroshima-u.ac.jp

BCC: ananya.rastogi@nature.com

Subject: Decision on Nature Computational Science manuscript NATCOMPUTSCI-22-0624A

Message: ** Please ensure you delete the link to your author homepage in this e-mail if you wish to forward it to your co-authors. **

Dear Professor Naoki,

Your manuscript "Decoding reward–curiosity conflict in decision-making from irrational behaviors" has now been seen by 2 referees, whose comments are appended below. You will see that while they find your work of interest, they have raised points that need to be addressed before we can make a decision on publication.

The referees' reports seem to be quite clear. Naturally, we will need you to address all of the points raised.

While we ask you to address all of the points raised, the following points need to be substantially worked on:

- Please show the equivalence between two types of free energy defined in line 475 and line 525.

In addition to these points, it would also be beneficial to address the following concerns:

- For the code, the link that has been provided is to the main user profile page on GitHub, and not to the repository. Please fix this.
- In addition, there is a lack of proper README files in the repository, and some files are in Japanese. Please edit these accordingly.

Please use the following link to submit your revised manuscript and a point-by-point response to the referees' comments (which should be in a separate document to any cover letter):

[REDACTED]

** This url links to your confidential homepage and associated information about manuscripts you may have submitted or be reviewing for us. If you wish to forward this e-mail to co-authors, please delete this link to your homepage first. **

To aid in the review process, we would appreciate it if you could also provide a copy of your manuscript files that indicates your revisions by making use of Track Changes or similar mark-up tools. Please also ensure that all correspondence is marked with your Nature Computational Science reference number in the subject line.

In addition, please make sure to upload a Word Document or LaTeX version of your text, to assist us in the editorial stage.

To improve transparency in authorship, we request that all authors identified as 'corresponding author' on published papers create and link their Open Researcher and Contributor Identifier (ORCID) with their account on the Manuscript Tracking System (MTS), prior to acceptance. ORCID helps the scientific community achieve unambiguous attribution of all scholarly contributions. You can create and link your ORCID from the home page of the MTS by clicking on 'Modify my Springer Nature account'. For more information please visit www.springernature.com/orcid.

We hope to receive your revised paper within two weeks. If you cannot send it within this time, please let us know.

Best regards,

Ananya Rastogi, PhD
Associate Editor
Nature Computational Science

Reviewers comments:

Reviewer #1 (Remarks to the Author):

Many thanks for responding to my previous observations – and congratulations on a thoughtful and challenging piece of work.

Reviewer #2 (Remarks to the Author):

[Major comments]

I greatly appreciate considerable effort have been made to address my concerns. While satisfied with most responses, I still have raised only two but critical concerns.

1. The authors presumably misunderstood Comments 2-7 by Reviewer 1 and Comment 1 by me (or FEP itself). The core of FEP is that inference (sensation from external world) and action selection (action to external world) can be explained as the minimization process of the same objective function, called "THE free energy". In this sense, FEP is often said to provide a "unified" theory of the brain. That is why I asked if the action selection rule (Eq. (3)) can be derived from the free energy (defined in line 475) in my Comment 1 (Otherwise, any free energy can be considered in adhoc manner.). Also, Comments 2-7 by Reviewer 1 suggested a technical hint to fill the gap between passive inference (hidden state estimation) and active inference (action selection) in the context of FEP, I guess. However, the authors have not followed Reviewer 1's suggestion or showed the equivalence between two types of free energy defined in line 475 and line 525, respectively. This is the most critical concern to be addressed before the publication if you would say that the proposed method is based on FEP (At lease, the authors should mention the gap as the current limitation though it is a compromise plan.).

2. The other concern is about the alternative version of the expected net utility defined in line 349. In principle, it's the same as the original one up to the proportional constant and only the different is between two prior distributions of c_t and d_t , which is caused by the variable transformation. Thus, I do not agree with the statement "which indicates that the rat suddenly no longer needed rewards. This should be unnatural for the animals that were starved before the experimental task for motivating to obtain food.", because the same thing is true for the original version, which underestimates the reward too much in spite of being in too starving condition. (I guess that the statement was added due to misunderstanding of concerns raised by the other reviewer.)

[Minor comments]

First of all, I would like to list up some errors or suspicious statements as follows.

- Line 272 (page 8): I wonder if " $y_{i,t}$ " should be " $p_{i,t}$ ", though I also missed it in

the first review. Please correct it if necessary.

- Line 281 (page 8): I wonder if " $\gamma_{i,t}$ " should be " $\sigma_{i,t}$ " in this context. Please correct it if necessary.
- Line 360 (page 10): " B_t " should be " β_t ", right? The errors appear several times in the same paragraph.
- Line 471 (page 13): In the first term of the right-hand, " \ln " should be taken over the whole fraction, but not separately for both the nominator and the denominator. I wonder if the correct form should be $\ln\{Q(w|\phi)/P(o,w|o,a)\}$ (the subscriptions are omitted for brief.). Please correct it if necessary.
- Line 522 (page 15): The temperature ($1/\beta$) is required before the second term of $\ln Q(a_{t+1})$.

Below is just an optional comment. I understood the superiority of the proposed iFEP over the Q-learning model in terms of the interpretation of internal state. But it would be great if the authors can compare the model fitness among three models presented in the paper using (approximate) marginal likelihood. If the result is better than or comparable with alternative model, the advantage of the proposed iFEP will be more reinforced.

Reviewer #2 (Remarks on code availability):

Actually, I cannot run several scripts in my environment due to my lack of System Identification Toolbox. However, all the programs look fine according to my visual inspection. README files are acceptable as well. So I have no request to improve the code.

Author Rebuttal, second revision:

Response to Editor's comments

Comment 1

Please show the equivalence between two types of free energy defined in line 475 and line 525.

In our response to the reviewer 2, we mentioned that there is a gap between two types of free energy and this gap we ventured to introduce is a core of irrational decision-making. Please see our response to comment 1.

Comment 2

For the code, the link that has been provided is to the main user profile page on GitHub, and not to the

repository. Please fix this.

Thank you for pointing it out. In revision, we provided a direct URL of the repository: https://github.com/YukiKonaka/Konaka_Honda_2023.

Comment 3

In addition, there is a lack of proper README files in the repository, and some files are in Japanese. Please edit these accordingly.

According to this suggestion, we uploaded README to the appropriate directory and deleted files containing Japanese.

Response to Reviewer 1's comments

Many thanks for responding to my previous observations – and congratulations on a thoughtful and challenging piece of work.

We are glad to hear that you evaluated our study

Response to Reviewer 2's comments

I greatly appreciate considerable effort have been made to address my concerns. While satisfied with most responses, I still have raised only two but critical concerns.

We have prepared the following responses to your valuable comments. We hope you will find them satisfactory.

Major comment 1

The authors presumably misunderstood Comments 2-7 by Reviewer 1 and Comment 1 by me (or FEP itself). The core of FEP is that inference (sensation from external world) and action selection (action to external world) can be explained as the minimization process of the same objective function, called “THE free energy”.

In this sense, FEP is often said to provide a “unified” theory of the brain. That is why I asked if the action selection rule (Eq. (3)) can be derived from the free energy (defined in line 475) in my Comment 1 (Otherwise, any free energy can be considered in adhoc manner.). Also, Comments 2-7 by Reviewer 1 suggested a technical hint to fill the gap between passive inference (hidden state estimation) and active inference (action selection) in the context of FEP, I guess. However, the authors have not followed Reviewer 1’s suggestion or showed the equivalence between two types of free energy defined in line 475 and line 525, respectively. This is the most critical concern to be addressed before the publication if you would say that the proposed method is based on FEP (At least, the authors should mention the gap as the current limitation though it is a compromise plan.).

Thank you for your accurate observation. In revision, we showed the derivation of the expected net utility at line 525 (line 520 in updated manuscript) from the free energy at line 475 (line 482 in updated manuscript). Please see lines from 525 to 568. In this derivation, however, there is the gap between the original EFP and our expected net utility. We think this gap appears when rational decision-making is assumed. On the other hand, we ventured to introduce a gap (i.e., the introduction of a curiosity parameter) by focusing on irrational decision-making. In revision, we mentioned our assumptions causing the gap (Lines 563-568).

Major comment 2

The other concern is about the alternative version of the expected net utility defined in line 349. In principle, it’s the same as the original one up to the proportional constant and only the different is between two prior distributions of c_t and d_t , which is caused by the variable transformation. Thus, I do not agree with the statement “which indicates that the rat suddenly no longer needed rewards. This should be unnatural for the animals that were starved before the experimental task for motivating to obtain food.”, because the same thing is true for the original version, which underestimates the reward too much in spite of being in too starving condition. (I guess that the statement was added due to misunderstanding of concerns raised by the other reviewer.)

> I do not agree with the statement “which indicates that the rat suddenly no longer needed rewards. This should be unnatural for the animals that were starved before the experimental task for motivating to obtain food.”, because ...

Sorry for our poor explanation. We think we cannot apply this statement to the actual rat. Thus, the alternative version of the model is not suited for describing the rat behavior. In revision, we clearly mentioned it (Lines 362).

> ... because the same thing is true for the original version, which underestimates the reward too much in spite of being in too starving condition.

Thank you for indicating the important point. In our estimation from the rat behaviors, the reward was estimated slightly smaller than the information gain (Fig. 6A-C). This seems very reasonable for a starved animal. This is because in starving conditions, animals desire to obtain more reward with higher confidence for survival. In revision, we clearly mentioned it (Lines 329-330).

Minor comment 1

- Line 272 (page 8): I wonder if “ $\gamma_{i,t}$ ” should be “ $p_{i,t}$ ”, though I also missed it in the first review. Please correct it if necessary.
- Line 281 (page 8): I wonder if “ $\gamma_{i,t}$ ” should be “ $o_{i,t}$ ” in this context. Please correct it if necessary.
- Line 360 (page 10): “ B_t ” should be “ β_t ”, right? The errors appear several times in the same paragraph.
- Line 471 (page 13): In the first term of the right-hand, “ln” should be taken over the whole fraction, but not separately for both the nominator and the denominator. I wonder if the correct form should be $\ln\{Q(w|\phi)/P(o,w|o,a)\}$ (the subscriptions are omitted for brief.). Please correct it if necessary.
- Line 522 (page 15): The temperature ($1/\beta$) is required before the second term of $\ln Q(a_{t+1})$.

Thank you for pointing out several mistakes. In revision, we corrected them.

Minor comment 2

Below is just an optional comment. I understood the superiority of the proposed iFEP over the Q-learning model in terms of the interpretation of internal state. But it would be great if the authors can compare the model fitness among three models presented in the paper using (approximate) marginal likelihood. If the result is better than or comparable with alternative model, the advantage of the proposed iFEP will be more reinforced.

As suggested, model comparison in terms of the fitness to the behavioral data is possible. However, we

think this does not add to the discussion, because three models (ReCU model and Q-learning model) have ability to fit the behavior by adjusting the time-dependent meta-parameters. In revision, we mentioned this point (Lines 425-428).

Minor comment 3 (Remarks on code availability):

Actually, I cannot run several scripts in my environment due to my lack of System Identification Toolbox. However, all the programs look fine according to my visual inspection. README files are acceptable as well. So I have no request to improve the code.

Thank you for checking our codes.

Decision Letter, second revision:

Date: 13th February 23 16:36:45
Last Sent: 13th February 23 16:36:45
Triggered By: Ananya Rastogi
From: ananya.rastogi@nature.com
To: nhonda@hiroshima-u.ac.jp
CC: computationalscience@nature.com
Subject: AIP Decision on Manuscript NATCOMPUTSCI-22-0624B
Message: Our ref: NATCOMPUTSCI-22-0624B

13th February 2023

Dear Dr. Naoki,

Thank you for submitting your revised manuscript "Decoding reward–curiosity conflict in decision-making from irrational behaviors" (NATCOMPUTSCI-22-0624B). It has now been seen by the original referees and their comments are below. The reviewers find that the paper has improved in revision, and therefore we'll be happy in principle to publish it in Nature Computational Science, pending minor revisions to satisfy the referees' final requests and to comply with our editorial and formatting guidelines.

TRANSPARENT PEER REVIEW

Nature Computational Science offers a transparent peer review option for original research manuscripts. We encourage increased transparency in peer review by publishing the reviewer comments, author rebuttal letters and editorial decision letters if the authors agree. Such peer review material is made available as a supplementary peer review file. **Please state in the cover letter 'I wish to participate in transparent peer review' if you want to opt in, or 'I do not wish to participate in transparent peer review' if you don't.** Failure to state your preference will result in delays in accepting your manuscript for publication. Please note: we allow redactions to authors' rebuttal and reviewer comments in the interest of confidentiality. If you are concerned about the release of confidential data, please let us know specifically what information you would like to have removed. Please note that we cannot incorporate redactions for any other reasons. Reviewer names will be published in the peer review files if the reviewer signed the comments to authors, or if reviewers explicitly agree to release their name. For more information, please refer to our [FAQ page](https://www.nature.com/documents/nr-transparent-peer-review.pdf).

Thank you again for your interest in Nature Computational Science Please do not hesitate to contact me if you have any questions.

Sincerely,

Ananya Rastogi, PhD
Associate Editor
Nature Computational Science

ORCID

Reviewer #2 (Remarks to the Author):

I appreciate the authors for their collaboration.
The presented manuscript has been revised adequately.
There is no more concern about the publication from me.

Reviewer #2 (Remarks on code availability):

Everything looks fine. I have no request to improve the code.

Final Decision Letter:

Date: 29th March 23 12:15:58

Last Sent: 29th March 23 12:15:58
Triggered By: Jie Pan
From: jie.pan@us.nature.com
To: nhonda@hiroshima-u.ac.jp
BCC: rjsproduction@springernature.com,fernando.chirigati@us.nature.com,jie.pan@us.nature.com,computationalscience@nature.com,rjsart@springernature.com
Subject: Decision on Nature Computational Science manuscript NATCOMPUTSCI-22-0624C
Message: Dear Professor Naoki,

We are pleased to inform you that your Article "Decoding reward–curiosity conflict in decision-making from irrational behaviors" has now been accepted for publication in Nature Computational Science.

Once your manuscript is typeset, you will receive an email with a link to choose the appropriate publishing options for your paper and our Author Services team will be in touch regarding any additional information that may be required.

Please note that Nature Computational Science is a Transformative Journal (TJ). Authors may publish their research with us through the traditional subscription access route or make their paper immediately open access through payment of an article-processing charge (APC). Authors will not be required to make a final decision about access to their article until it has been accepted. Find out more about Transformative Journals

Authors may need to take specific actions to achieve compliance with funder and institutional open access mandates. If your research is supported by a funder that requires immediate open access (e.g. according to Plan S principles) then you should select the gold OA route, and we will direct you to the compliant route where possible. For authors selecting the subscription publication route, the journal's standard licensing terms will need to be accepted, including self-archiving policies. Those licensing terms will supersede any other terms that the author or any third party may assert apply to any version of the manuscript.

Acceptance of your manuscript is conditional on all authors' agreement with our publication policies (see <https://www.nature.com/natcomputsci/for-authors>). In particular your manuscript must not be published elsewhere and there must be no announcement of the work to any media outlet until the publication date (the day on which it is uploaded onto

our web site).

Before your manuscript is typeset, we will edit the text to ensure it is intelligible to our wide readership and conforms to house style. We look particularly carefully at the titles of all papers to ensure that they are relatively brief and understandable.

Once your manuscript is typeset and you have completed the appropriate grant of rights, you will receive a link to your electronic proof via email with a request to make any corrections within 48 hours. If, when you receive your proof, you cannot meet this deadline, please inform us at rjsproduction@springernature.com immediately.

If you have queries at any point during the production process then please contact the production team at rjsproduction@springernature.com. Once your paper has been scheduled for online publication, the Nature press office will be in touch to confirm the details.

Content is published online weekly on Mondays and Thursdays, and the embargo is set at 16:00 London time (GMT)/11:00 am US Eastern time (EST) on the day of publication. If you need to know the exact publication date or when the news embargo will be lifted, please contact our press office after you have submitted your proof corrections. Now is the time to inform your Public Relations or Press Office about your paper, as they might be interested in promoting its publication. This will allow them time to prepare an accurate and satisfactory press release. Include your manuscript tracking number NATCOMPUTSCI-22-0624C and the name of the journal, which they will need when they contact our office.

About one week before your paper is published online, we shall be distributing a press release to news organizations worldwide, which may include details of your work. We are happy for your institution or funding agency to prepare its own press release, but it must mention the embargo date and Nature Computational Science. Our Press Office will contact you closer to the time of publication, but if you or your Press Office have any inquiries in the meantime, please contact press@nature.com.

We welcome the submission of potential cover material (including a short caption of around 40 words) related to your manuscript; suggestions should be sent to Nature Computational Science as electronic files (the image should be 300 dpi at 210 x 297 mm in either TIFF or JPEG format). We also welcome suggestions for the Hero Image, which appears at the top of our [home page](http://www.nature.com/natcomputsci); these should be 72 dpi at 1400 x 400 pixels in JPEG format. Please note that such pictures should be selected more for their aesthetic appeal than for their scientific content, and that colour images work better than black and white or grayscale images. Please do not try to design a cover with the Nature Computational Science logo etc., and please do not submit composites of images related to your work. I am sure you will understand that we cannot make any promise as to whether any of your suggestions might be selected for the cover of the journal.

Best regards,

Jie Pan, Ph.D.
Senior Editor
Nature Computational Science

On behalf of:

Ananya Rastogi, PhD
Associate Editor
Nature Computational Science

P.S. Click on the following link if you would like to recommend Nature Computational Science to your librarian: https://www.springernature.com/gp/librarians/recommend-to-your-library

** Visit the Springer Nature Editorial and Publishing website at www.springernature.com/editorial-and-publishing-jobs for more information about our career opportunities. If you have any questions please click here.**